# Bartlett-corrected tests for varying precision beta regressions with application to environmental biometrics

**Ana C. Guedes, Francisco Cribari-Neto, Patrícia L. Espinheira**👁 *

Departamento de Estatística, Universidade Federal de Pernambuco, Recife, PE, Brazil

* patespipa@de.ufpe.br

**Data Availability Statement:** The data and computer code are available at https://github.com/acguedes/beta-Bartlett.

**Funding:** The first author of this paper is a PhD student in the PosGraduate Program in Statistics of

## Abstract

Beta regressions are commonly used with responses that assume values in the standard unit interval, such as rates, proportions and concentration indices. Hypothesis testing inferences on the model parameters are typically performed using the likelihood ratio test. It delivers accurate inferences when the sample size is large, but can otherwise lead to unreliable conclusions. It is thus important to develop alternative tests with superior finite sample behavior. We derive the Bartlett correction to the likelihood ratio test under the more general formulation of the beta regression model, i.e. under varying precision. The model contains two submodels, one for the mean response and a separate one for the precision parameter. Our interest lies in performing testing inferences on the parameters that index both submodels. We use three Bartlett-corrected likelihood ratio test statistics that are expected to yield superior performance when the sample size is small. We present Monte Carlo simulation evidence on the finite sample behavior of the Bartlett-corrected tests relative to the standard likelihood ratio test and to two improved tests that are based on an alternative approach. The numerical evidence shows that one of the Bartlett-corrected typically delivers accurate inferences even when the sample is quite small. An empirical application related to behavioral biometrics is presented and discussed.

## Introduction

Regression models are useful for gaining knowledge on how different variables (known as regressors, covariates or independent variables) impact the mean behavior of a variable of interest (known as dependent variable or response). The beta regression model is the most commonly used model with responses that are double bounded, in particular with responses that assume values in the standard unit interval, $(0, 1)$. It was introduced by [1] who used an alternative parameterization for the beta density, which is indexed by mean ($\mu$) and precision ($\phi$) parameters. Let $Y$ be a beta-distributed random variable. Its density is

$$b(y; \mu, \phi) = \frac{\Gamma(\phi)}{\Gamma(\mu\phi)\Gamma(\phi(1-\mu))} y^{\mu\phi-1}(1-y)^{\phi(1-\mu)-1}, \quad 0 < y < 1, \tag{1}$$

the Federal University of Pernambuco. Aiming dedicate herself exclusively to the doctorate whitout to must work for her support, purchase of books, rent of dormitories. The Conselho Nacional de Desenvolvimento e pesquisa (CNPq) provides a monthly financial aid to the student in the amount of U$405.00. The student comes from the state of Ceará, northeast of Brazil, admittedly poor, and her parents don't have the minimum conditions of subsistence, let alone to give the education of her daughter's doctorate level. The funders had no role in study design, data collection and analysis, decision to publish, or preparation of the manuscript.

**Competing interests:** The authors have declared that no competing interests exist.

$0 < \mu < 1, \phi > 0$, where $\Gamma(\cdot)$ is the gamma function. Such a law is quite flexible in the sense that the density in (1) can assume different shapes depending on the parameter values. It was used by [1] as the underlying foundation for a regression model in which $y_1, \ldots, y_n$ are independent random variables such that $y_i$ is beta-distributed with mean $\mu_i$ (i.e. $\mathbb{E}(y_i) = \mu_i$) and precision parameter $\phi$, for $i = 1, \ldots, n$. They showed that the variance of $y_i$ is $\mu_i(1 - \mu_i)/(1 + \phi)$ which, for a given $\mu_i$, is decreasing in $\phi$. The model is thus heteroskedastic since the variance of $y_i$ changes with $\mu_i$. The response means are modeled using a set of covariates and $\phi$ is assumed constant across observations. This model became known as the fixed precision beta regression model.

A more general beta regression formulation was considered by [2] and formally introduced by [3] who allowed the precision parameter to vary across observations, i.e. $y_i$ is beta-distributed with mean $\mu_i$ and precision $\phi_i$, $i = 1, \ldots, n$. More flexibility can be achieved in some situations by allowing the precision parameter to be impacted by some covariate values. In such a more general formulation, the variance of $y_i$ is no longer restricted to be a multiple of $\mu_i(1 - \mu_i)$. The model includes two separate regression submodels, one for the mean and another for the precision, and became known as the variable precision beta regression model. The fixed precision beta regression model is a particular case of the variable precision counterpart; it is obtained by setting $\phi_1 = \cdots = \phi_n = \phi$.

Fixed and varying precision beta regression modeling have been used in many different fields. A beta regression analysis of the effects of sexual maturity on space use in Atlantic salmon (Salmo salar) parr can be found in [4]. In [5] the beta regression model is used to segment and describe the container shipping market by analyzing the relationships between service attributes and likelihood of customer retention for the container shipping industry. Some applications of beta regression modeling in ecology can be found in [6]. In [7], a statistical downscaling model is developed based on beta regression which allow precipitation state in river basin to be calculated. The beta regression model is used by [8] to model global solar radiation. For a beta regression analysis of ischemic stroke volume, see [9].

In both variants of the beta regression model (fixed and variable precision), parameter estimation is carried out by maximum likelihood. It is common practice to perform testing inferences via likelihood ratio and $z$-tests. The latter are Wald-type tests and are typically less accurate than the former; see [10]. Point estimation and testing inferences are usually accurate when the sample size ($n$) is large. In some applications, nonetheless, the number of data points is small and it is recommended to make use of inferential tools that are expected to yield reliable inferences in small samples. For instance [11], obtained modified parameter estimates that display smaller biases in fixed and variable precision linear beta regression models.

The likelihood ratio test, which is commonly used in beta regression empirical analyses, employs an asymptotic approximation: the critical values used in the test are obtained from the test statistic's asymptotic null distribution, which is known to be $\chi_l^2$, where $l$ is the number of restrictions under evaluation. An asymptotic approximation is used because the test statistic's exact null distribution is unknown. In large samples, the test typically delivers accurate inferences since there is good control of the type I error frequency. In contrast, when the number of data points is small, size distortions can be large. In particular, the test tends to be liberal (oversized): the effective null rejection rates tend to be considerably larger than the selected significance level. When the sample size is quite small, the test's effective null rejection can be much larger than the nominal significance level, as shown by the numerical evidence we report. A Bartlett correction to the likelihood ratio test was derived by [12]. A major shortcoming of their result, however, is that it only holds for the fixed precision beta regression model. In this paper, we overcome such a shortcoming by deriving the Bartlett correction for varying precision beta regressions, which are more commonly used by practitioners. The derivation of

the correction becomes more challenging in the more general setting. That happens because the parameters that index the two submodels are not orthogonal in the sense that Fisher's information matrix is not block diagonal, and that renders lengthier and more complex derivations of the quantities involved in the Bartlett correction. We considered three Bartlett-corrected test statistics. It is noteworthy that the size distortions of such tests vanish faster than those of the standard likelihood ratio test as the sample size increases and thus the new tests are expected to outperform the likelihood ratio test in small samples. In particular, the likelihood ratio test's size distortions are $O(n^{-1})$ whereas those of the Bartlett-corrected tests are $O(n^{-2})$.

To motivate our analysis, consider the following important issue in behavioral biometrics: the impact of average intelligence on the prevalence of religious disbelievers. Suppose there is interest in measuring such a net impact using data on $n$ nations. The variable of interest (response) is the proportion of atheists in each country and the covariates include average intelligence and other control variables. [13] carried out varying precision beta regression analyses and produced estimates of such an impact under different scenarios. Each scenario corresponds to a particular choice of countries. We consider the scenario that uses data on 50 countries. We show that by using corrected likelihood ratio tests we arrive at a varying precision beta regression model different from that used by the authors. It is noteworthy that our model yields a better fit than their model. We also note that the maximal estimated impact of intelligence on religious disbelief obtained from our model is considerably larger than that computed from the model in [13] in low income nations. Our results also reveal that, as countries become more developed, the maximal impact of intelligence on the prevalence of atheists weakens and the impact becomes, in the plausible range of average intelligence values, more symmetric. To the best of our knowledge, this is the first analysis of how the maximal impact of average intelligence on the prevalence of atheists is affected by economic development. This illustrates the importance of using tests with good small sample performance when performing beta regression analyses with samples of small to moderate sizes.

The remainder of the paper is structured as follows. In first section that follows this introduction, we present the variable precision beta regression model. In the second section, we derive the Bartlett correction to the likelihood ratio test in varying precision beta regressions and use it in three modified test statistics. Our main contribution is that we obtain closed-form expressions for the quantities that allow improved testing inferences to be carried out in varying precision beta regressions. Additionally, we briefly review an alternative small sample correction that is already available in the literature. Unlike the correction we derive, however, it does not yield an improvement in the rate at which size distortions vanish. In particular, the size distortions of our corrected tests vanish at rate $O(n^{-2})$ whereas those of the alternative tests we consider do so at rate $O(n^{-1})$. Monte Carlo simulation evidence is presented in the third section. An empirical application that addresses an important issue in behavioral biometrics is presented and discussed in the fourth section. The fifth section contains some concluding remarks. Technical details related to the derivation of the quantities involved in the Bartlett correction are presented in the Appendix.

## The beta regression model

Let $y = (y_1, \ldots, y_n)^\top$ be a vector of independent random variables such that $y_i$ follows the beta distribution with mean $\mu_i$ and precision $\phi_i$, $i = 1, \ldots, n$. Such parameters are modeled as

$$g_1(\mu_i) = \eta_i = \sum_{j=1}^{p} \beta_j x_{ij} \quad \text{and} \quad g_2(\phi_i) = \zeta_i = \sum_{j=1}^{q} \delta_j h_{ij},$$

where $\beta = (\beta_1, \ldots, \beta_p)^\top \in \mathrm{IR}^p$ and $\delta = (\delta_1, \ldots, \delta_q)^\top \in \mathrm{IR}^q$ are unknown regression parameters ($p + q < n$), $\eta_i$ and $\zeta_i$ are linear predictors, $x_{i1} \equiv h_{i1} \equiv 1 \forall i$, $x_{i2}, \ldots, x_{ip}$ and $h_{i2}, \ldots, h_{iq}$ are mean and precision covariates, respectively, and $g_1 : (0, 1) \mapsto \mathrm{IR}$ and $g_2 : (0, \infty) \mapsto \mathrm{IR}$ are strictly monotonic and twice-differentiable link functions. Common choices for $g_1$ are logit, probit, loglog, cloglog and Cauchy, and common choices for $g_2$ are log and square root; see [14].

Let $\theta = (\beta^\top, \delta^\top)^\top$ be the vector containing all regression coefficients. The log-likelihood function is

$$\ell(\theta) \equiv \ell(\beta, \delta) = \sum_{i=1}^{n} \ell_i(\mu_i, \phi_i), \tag{2}$$

where $\ell_i(\mu_i, \phi_i) = \log\Gamma(\phi_i) - \log\Gamma(\mu_i\phi_i) - \log\Gamma((1 - \mu_i)\phi_i) + (\mu_i\phi_i - 1)y_i^* + (\phi_i - 2)y_i^\dagger$, with $y_i^* = \log(y_i/1 - y_i)$ and $y_i^\dagger = \log(1 - y_i)$. The maximum likelihood estimators of $\beta$ and $\delta$ solve $U = \partial\ell(\beta, \delta)/\partial\theta = (U_\beta(\beta, \delta)^\top, U_\delta(\beta, \delta)^\top)^\top = 0_{p+q}$, where $0_{p+q}$ is a $(p+q)$-vector of zeros. They cannot be expressed in closed-form. Maximum likelihood estimates can be obtained by numerically maximizing the model log-likelihood function using a Newton or quasi-Newton optimization algorithm such as the Broyden-Fletcher-Goldfarb-Shanno (BFGS) algorithm; see [15].

For a recent overview of the beta regression model, see [6]. Practitioners can perform beta regression analyses using the `betareg` package developed for the `R` statistical computing environment; see [14].

## Improved likelihood ratio tests in beta regressions

At the outset, we consider a general setup. Suppose the interest lies in testing a null hypothesis ($\mathcal{H}_0$) that imposes $l$ restrictions on the $k$-dimensional parameter vector $\theta = (\beta^\top, \delta^\top)^\top$, where $k = p + q$. To that end, we write $\theta = (\psi^\top, \lambda^\top)^\top$, where $\psi = (\psi_1, \ldots, \psi_l)^\top$ is the vector of parameters of interest and $\lambda = (\lambda_1, \ldots, \lambda_s)^\top$ is the vector of nuisance parameters so that $l + s = p + q$. We wish to test $\mathcal{H}_0 : \psi = \psi^{(0)}$ against $\mathcal{H}_1 : \psi \neq \psi^{(0)}$, where $\psi^{(0)}$ is a given $l$-vector. The likelihood ratio test statistic is

$$\omega = 2[\ell(\hat{\psi}, \hat{\lambda}) - \ell(\psi^0, \tilde{\lambda})],$$

where $(\hat{\psi}^\top, \hat{\lambda}^\top)$ and $(\psi^{0\top}, \tilde{\lambda}^\top)$ are the unrestricted and restricted maximum likelihood estimators of $(\psi^\top, \lambda^\top)$, respectively. Under the null hypothesis, $w$ is asymptotically distributed as $\chi_l^2$. The test is usually performed using critical values obtained from such an asymptotic null distribution, the approximation error being of order $O(n^{-1})$. That is, under the null hypothesis, $\Pr(\omega > \chi_{l;1-\alpha}^2) = \alpha + O(n^{-1})$, where $\alpha \in (0, 1)$ is the test significance level and $\chi_{l;1-\alpha}^2$ is the $(1 - \alpha)$th quantile from the $\chi_l^2$ distribution. The chi-squared approximation to the null distribution of $\omega$ may be poor when the sample size is small and, as a result, large size distortions may take place.

A correction that became known as 'the Bartlett correction' was developed to improve the likelihood ratio test's small sample behavior. It uses the fact that, under $\mathcal{H}_0$, $\mathrm{IE}(\omega) = l + b + O(n^{-2})$, where $b = b(\theta)$ is $O(n^{-1})$. Using such a result, it is possible to define the corrected test statistic

$$\omega_{b1} = \frac{\omega}{1 + b/l}$$

whose expected value equals $l$ when terms of order $O(n^{-2})$ are neglected. The quantity $c = 1 + b/l$ became known as 'the Bartlett correction factor'. A general approach for obtaining the

Bartlett correction factor in statistical models was developed by [16]. His approach requires the derivation of log-likelihood cumulants. The expected value of $\omega$, under the null hypothesis, can be expressed as

$$\mathbb{E}(\omega) = l + \varepsilon_k - \varepsilon_{k-l} + O(n^{-2}),$$

where $\varepsilon_k$ and $\varepsilon_{k-l}$ are of order $O(n^{-1})$. Here,

$$\varepsilon_k = \sum_\theta (\lambda_{rstu} - \lambda_{rstuvw}),\tag{3}$$

where

$$\lambda_{rstu} = \kappa^{rs}\kappa^{tu}\left\{\frac{\kappa_{rstu}}{4} - \kappa_{rst}^{(u)} + \kappa_{rt}^{(su)}\right\} \text{ and } \lambda_{rstuvw} = \kappa^{rs}\kappa^{tu}\kappa^{vw}\left\{\kappa_{rtv}\left(\frac{\kappa_{suw}}{6}\right.\right.$$
$$\left.\left. -\kappa_{sw}^{(u)}\right) + \kappa_{rtu}\left(\frac{\kappa_{svw}}{4} - \kappa_{sw}^{(v)}\right) + \kappa_{rt}^{(v)}\kappa_{sw}^{(u)} + \kappa_{rt}^{(u)}\kappa_{sw}^{(v)}\right\}.$$

The above cumulants ($\kappa$'s) are defined in the Appendix. The indices $r$, $s$, $t$, $u$, $v$ and $w$ vary over all $k$ parameters in the summation in (3). The Bartlett correction factor can then be written as

$$c = 1 + \frac{\varepsilon_k - \varepsilon_{k-l}}{l}.$$

[16] also showed that all cumulants of the Bartlett-corrected test statistic agree with those of the reference chi-squared distribution with error of order $O(n^{-3/2})$ which indicates that its null distribution is expected to be well approximated by the limiting chi-squared distribution. [17] obtained an asymptotic expansion for the null distribution of $\omega$; see also [18–20]. [21] showed that size distortions of Bartlett-corrected tests are of order $O(n^{-2})$, and not of order $O(n^{-3/2})$, as previously believed.

In what follows, we shall obtain the Bartlett correction factor for the class of varying precision beta regressions. We shall only present the main result. Details on the derivation can be found in the Appendix. It is noteworthy that $\beta$ and $\delta$ are not orthogonal (i.e., Fisher's information matrix is not block diagonal), unlike what happens in the class of generalized linear models. As a consequence, the derivation of the Bartlett correction factor becomes lengthier and more challenging. We shall use the main result in [22], who wrote the general adjustment factor in matrix form. At the outset, we define some $k \times k$ matrices whose $(r, s)$ elements are

$$A^{(tu)} = \left\{\frac{\kappa_{rstu}}{4} - \kappa_{rst}^{(u)} + \kappa_{rt}^{(su)}\right\}, \quad P^{(t)} = \{\kappa_{rst}\}, \quad Q^{(u)} = \{\kappa_{su}^{(r)}\},$$

$t, u = 1, \ldots, k$. We derived the log-likelihood cumulants up to fourth order for the class of varying precision beta regression models. These cumulants are presented in the Appendix. Using such results, we obtain matrices $A^{(tu)}$, $P^{(t)}$ and $Q^{(u)}$. It is then possible to write $\varepsilon_k$ as

$$\varepsilon_k = \text{tr}(K^{-1}(L - M - N)),\tag{4}$$

where $\mathrm{tr}(\cdot)$ is the trace operator and the $(r, s)$ elements of $L$, $M$ and $N$ are

$$L_{rs} = \{\mathrm{tr}(K^{-1}A^{(rs)})\},$$

$$M_{rs} = -\frac{1}{6}\{\mathrm{tr}(K^{-1}P^{(r)}K^{-1}P^{(s)})\} + \{\mathrm{tr}(K^{-1}P^{(r)}K^{-1}Q^{(s)\top})\}$$
$$- \{\mathrm{tr}(K^{-1}Q^{(r)}K^{-1}Q^{(s)})\},$$

$$N_{rs} = -\frac{1}{4}\{\mathrm{tr}(P^{(r)}K^{-1})\mathrm{tr}(P^{(s)}K^{-1})\} + \{\mathrm{tr}(P^{(r)}K^{-1})\mathrm{tr}(Q^{(s)}K^{-1})\}$$
$$- \{\mathrm{tr}(Q^{(r)}K^{-1})\mathrm{tr}(Q^{(s)}K^{-1})\},$$

$r, s = 1, \ldots, k$. Also, $\varepsilon_{k-l}$ is obtained from (4) by only considering the nuisance parameters.

The corrected statistic $\omega_{b1}$ is the standard Bartlett-corrected likelihood ratio test statistic. In addition to it, we shall also consider two other Bartlett-corrected test statistics that are used in [23]. The three test statistics are equivalent up to order $O(n^{-1})$ and are given by

$$\omega_{b1} = \frac{\omega}{c}, \quad \omega_{b2} = \omega\exp\left\{-\frac{(\varepsilon_k - \varepsilon_{k-l})}{l}\right\}, \quad \omega_{b3} = \omega\left\{1 - \frac{(\varepsilon_k - \varepsilon_{k-l})}{l}\right\}.$$

We shall refer to the three corrected test statistics above as 'ratio-like', 'exponentially adjusted' and 'multiplicative-like', respectively. An advantage of $\omega_{b2}$ is that it is always positive-valued. In order to use the above test statistics in a given class of models, it is necessary to obtain closed-form expressions for $\varepsilon_k$ and $\varepsilon_{k-l}$ that are valid for such models. For varying precision beta regressions, these quantities can be computed using Eq (4), which is our main result. For details on Bartlett corrections, we refer readers to [24, 25].

An alternative correction to the likelihood ratio test statistic was proposed by [26] who generalized previous results in [27]. His main result relate to those in [28, 29]. The author in [26] proposed using the following two modified test statistics: $\omega_{a1} = \omega - 2\log\xi$ and $\omega_{a2} = \omega(1 - \omega^{-1}\log\xi)^2$, the latter having the advantage of always being positive-valued. $\xi$ is a function of several model-based quantities (score function, expected information, observed information, etc.). Closed-form expressions for $\xi$ were derived by several authors considering different underlying models. In particular, for models tailored for double limited responses, they were derived by [30] for unit gamma regressions, by [31] for varying precision regression models, and by [32] for beta regressions with parametric mean link function. The finite sample performances of such corrected tests when used in beta regressions was numerically evaluated by [10].

It is noteworthy that the size distortions of the three Bartlett-corrected tests vanish at a faster rate than those of $\omega$, $\omega_{a1}$ and $\omega_{a2}$ as the sample size increases: $O(n^{-2})$ versus $O(n^{-1})$.

Finally, we note that there are alternative strategies for achieving accurate hypothesis testing inferences in small samples. For instance [33], proposed a numerical approach for estimating the Bartlett correction factor and [34] obtained the Bartlett correction for generalized linear models using a modified version of the likelihood function that accounts for the impact of nuisance parameters on the inference made on the parameters of interest. We shall not pursue these approaches since, as we shall see, the standard Bartlett corrected test is able to deliver extremely accurate inference in small samples in varying precision beta regressions even when the number of nuisance parameters is large.

## Numerical evidence

In what follows we shall present Monte Carlo simulation results on the finite sample performances of six tests in varying precision beta regressions, namely: $\omega$, $\omega_{b1}$ ('ratio-like'), $\omega_{b2}$ ('exponentially adjusted'), $\omega_{b3}$ ('multiplicative-like'), $\omega_{a1}$ and $\omega_{a2}$. All reported results are

**Table 1. Null rejection rates (%), $\mathcal{H}_0: \boldsymbol{\beta_4 = 0}$.**

| | $\alpha = 10\%$ | | | | $\alpha = 5\%$ | | | | $\alpha = 1\%$ | | | |
|---|---|---|---|---|---|---|---|---|---|---|---|---|
| | *n* | | | | *n* | | | | *n* | | | |
| | 15 | 20 | 30 | 40 | 15 | 20 | 30 | 40 | 15 | 20 | 30 | 40 |
| $\omega$ | 30.1 | 25.2 | 19.2 | 15.7 | 21.8 | 17.6 | 12.1 | 9.0 | 10.7 | 7.4 | 4.0 | 2.9 |
| $\omega_{b1}$ | 19.5 | 16.7 | 12.8 | 11.1 | 11.8 | 9.8 | 6.9 | 5.8 | 3.8 | 2.8 | 1.5 | 1.2 |
| $\omega_{b2}$ | 16.6 | 14.6 | 11.6 | 10.6 | 9.5 | 8.2 | 6.0 | 5.4 | 2.4 | 1.8 | 1.1 | 1.1 |
| $\omega_{b3}$ | 8.2 | 10.4 | 10.0 | 9.9 | 3.5 | 4.9 | 4.6 | 5.0 | 0.4 | 0.8 | 0.7 | 0.9 |
| $\omega_{a1}$ | 16.0 | 14.7 | 10.6 | 10.0 | 10.2 | 9.3 | 5.3 | 5.2 | 4.0 | 3.0 | 1.2 | 1.1 |
| $\omega_{a2}$ | 19.4 | 17.2 | 11.5 | 10.4 | 12.9 | 11.1 | 5.9 | 5.5 | 5.6 | 3.9 | 1.4 | 1.2 |

based on 10,000 replications and were obtained using the R statistical computing environment; see [35]. Log-likelihood maximization was performed using the Broyden-Fletcher-Goldfarb-Shanno (BFGS) algorithm with analytical first derivatives. Starting values for $\beta$ and $\delta$ were computed as described in Appendix A of [36] with minor tweaks. The computation of such starting values entails the estimation of two linear regressions. We consider the varying precision beta regression model $\log(\mu_i/(1 - \mu_i) = \beta_1 + \beta_2 x_{i2} + \beta_3 x_{i3} + \beta_4 x_{i4}$ and $\log(\phi_i) = \delta_1 + \delta_2 h_{i2} + \delta_3 h_{i3}$, $i = 1, \ldots, n$. All covariate values were obtained as $\mathcal{U}(-0.5, 0.5)$ random draws and remained constant for all replications performed for a given sample size. We consider three scenarios. In the first scenario, we test $\mathcal{H}_0 : \beta_4 = 0$, and hence $l = 1$ (one restriction). The true parameter values are $\beta_1 = 1.0$, $\beta_2 = 1.7$, $\beta_3 = 3.5$, $\beta_4 = 0$, $\delta_1 = 3.7$, $\delta_2 = 1.5$ and $\delta_3 = 0.9$. In the second scenario, the interest lies in testing $\mathcal{H}_0: \beta_3 = \beta_4 = 0$, thus $l = 2$ (two restrictions). The parameter values in this case are $\beta_1 = 1.0$, $\beta_2 = 1.7$, $\beta_3 = \beta_4 = 0$, $\delta_1 = 3.7$, $\delta_2 = 1.5$ and $\delta_3 = 0.9$. In the third and final scenario, the null hypothesis under evaluation is $\mathcal{H}_0: \delta_2 = \delta_3 = 0$, and hence $l = 2$ (two restrictions). The parameter values are $\beta_1 = 1.0$, $\beta_2 = 1.7$, $\beta_3 = 2.5$, $\beta_4 = -3.0$, $\delta_1 = 3.7$ and $\delta_2 = \delta_3 = 0$. We computed the tests' null rejection rates at the $\alpha = 10\%$, 5%, 1% significance levels for different sample sizes ($n \in \{15, 20, 30, 40\}$). They are presented in Table 1 (first scenario), Table 2 (second scenario) and Table 3 (third scenario); all entries are percentages.

The tests' null rejection rates for the first scenario are, as noted, displayed in Table 1. At the outset, we note that the likelihood ratio test $\omega$ is considerably liberal, that is, it rejects the null hypothesis too often when it is true. For instance, when $n = 15$ and $\alpha = 10\%$, its null rejection rate exceeds 30%, i.e. it is over three time larger than the nominal significance level. When $n = 20$, it equals 25.2%. The test is considerably oversized even when $n = 40$ (null rejection rate > 15%). The corrected tests display much better control of the type I error frequency, especially the third Bartlett-corrected test (i.e. that based on $\omega_{b3}$—'multiplicative-like'). For example, when $n = 20$ and $\alpha = 10\%$, its null rejection is 10.4% whereas those of $\omega_{b1}$, $\omega_{b2}$, $\omega_{a1}$

**Table 2. Null rejection rates (%), $\mathcal{H}_0 : \beta_3 = \beta_4 = 0$.**

| | $\alpha = 10\%$ | | | | $\alpha = 5\%$ | | | | $\alpha = 1\%$ | | | |
|---|---|---|---|---|---|---|---|---|---|---|---|---|
| | *n* | | | | *n* | | | | *n* | | | |
| | 15 | 20 | 30 | 40 | 15 | 20 | 30 | 40 | 15 | 20 | 30 | 40 |
| $\omega$ | 39.6 | 31.1 | 21.2 | 16.6 | 29.9 | 22.0 | 12.9 | 9.5 | 16.5 | 9.6 | 4.2 | 2.8 |
| $\omega_{b1}$ | 23.4 | 18.2 | 12.6 | 10.9 | 15.2 | 10.7 | 6.7 | 5.6 | 5.5 | 2.7 | 1.5 | 1.1 |
| $\omega_{b2}$ | 19.1 | 15.7 | 11.5 | 10.5 | 11.4 | 8.6 | 5.9 | 5.2 | 3.3 | 1.8 | 1.2 | 1.0 |
| $\omega_{b3}$ | 7.4 | 10.7 | 9.9 | 9.6 | 2.9 | 4.7 | 4.8 | 4.7 | 0.4 | 0.7 | 0.9 | 0.8 |
| $\omega_{a1}$ | 16.7 | 14.9 | 10.5 | 9.8 | 10.8 | 8.8 | 5.3 | 4.8 | 4.1 | 2.7 | 1.1 | 0.9 |
| $\omega_{a2}$ | 21.6 | 17.5 | 11.3 | 10.3 | 14.1 | 10.7 | 6.0 | 5.1 | 5.8 | 3.4 | 1.3 | 1.0 |

**Table 3. Null rejection rates (%), $\mathcal{H}_0 : \delta_2 = \delta_3 = 0$.**

| | $\alpha = 10\%$ | | | | $\alpha = 5\%$ | | | | $\alpha = 1\%$ | | | |
|---|---|---|---|---|---|---|---|---|---|---|---|---|
| | $n$ | | | | $n$ | | | | $n$ | | | |
| | 15 | 20 | 30 | 40 | 15 | 20 | 30 | 40 | 15 | 20 | 30 | 40 |
| $\omega$ | 39.6 | 29.8 | 20.3 | 16.3 | 30.2 | 20.7 | 12.9 | 9.3 | 15.9 | 8.8 | 4.4 | 2.7 |
| $\omega_{b1}$ | 23.6 | 17.1 | 12.9 | 11.0 | 14.7 | 9.7 | 6.8 | 5.3 | 4.8 | 2.4 | 1.6 | 1.1 |
| $\omega_{b2}$ | 19.1 | 14.4 | 12.0 | 10.3 | 10.9 | 7.5 | 6.2 | 5.0 | 2.7 | 1.8 | 1.3 | 1.0 |
| $\omega_{b3}$ | 7.7 | 9.5 | 10.1 | 9.7 | 3.3 | 4.1 | 5.0 | 4.7 | 0.4 | 0.7 | 0.8 | 0.8 |
| $\omega_{a1}$ | 13.1 | 11.7 | 10.3 | 9.7 | 7.4 | 6.1 | 5.3 | 4.8 | 1.9 | 1.5 | 1.2 | 1.0 |
| $\omega_{a2}$ | 18.0 | 14.1 | 11.3 | 10.2 | 10.8 | 7.7 | 5.9 | 5.0 | 3.3 | 2.0 | 1.4 | 1.0 |

and $\omega_{a2}$, are, respectively, 16.7%, 14.6%, 14.7% and 17.2%. All modified tests display small size distortions when $n = 40$; again, $\omega_{b3}$ ('mutiplicative-like') is the best performer. Interestingly, $\omega_{b3}$ is the only conservative test when the sample size is very small ($n = 15$).

Fig 1 contains quantile-quantile (QQ) plots of three test statistics, namely: the likelihood ratio test statistic, the best performing Bartlett-corrected test statistic ($\omega_{b3}$—'mutiplicative-like') and the best performing test statistics obtained from the alternative finite sample correction ($\omega_{a1}$). We plot the exact quantiles of the three test statistics against their asymptotic counterparts (obtained from the $\chi_1^2$ distribution). The included 45° line indicates perfect agreement between exact and asymptotic null distributions. The left and right panels are for $n = 15$ and $n = 20$, respectively. In both plots, the line that corresponds to $\omega$ is considerably above the 45° line which indicates that the test statistic exact quantiles are much larger than the asymptotic quantiles, and that translates into liberal test behavior, i.e., the test tends to overreject the null hypothesis. The exact quantiles of $\omega_{a1}$ also exceed those from the chi-squared distribution, but less dramatically. The null distribution of $\omega_{b3}$, the Bartlett-corrected test statistic

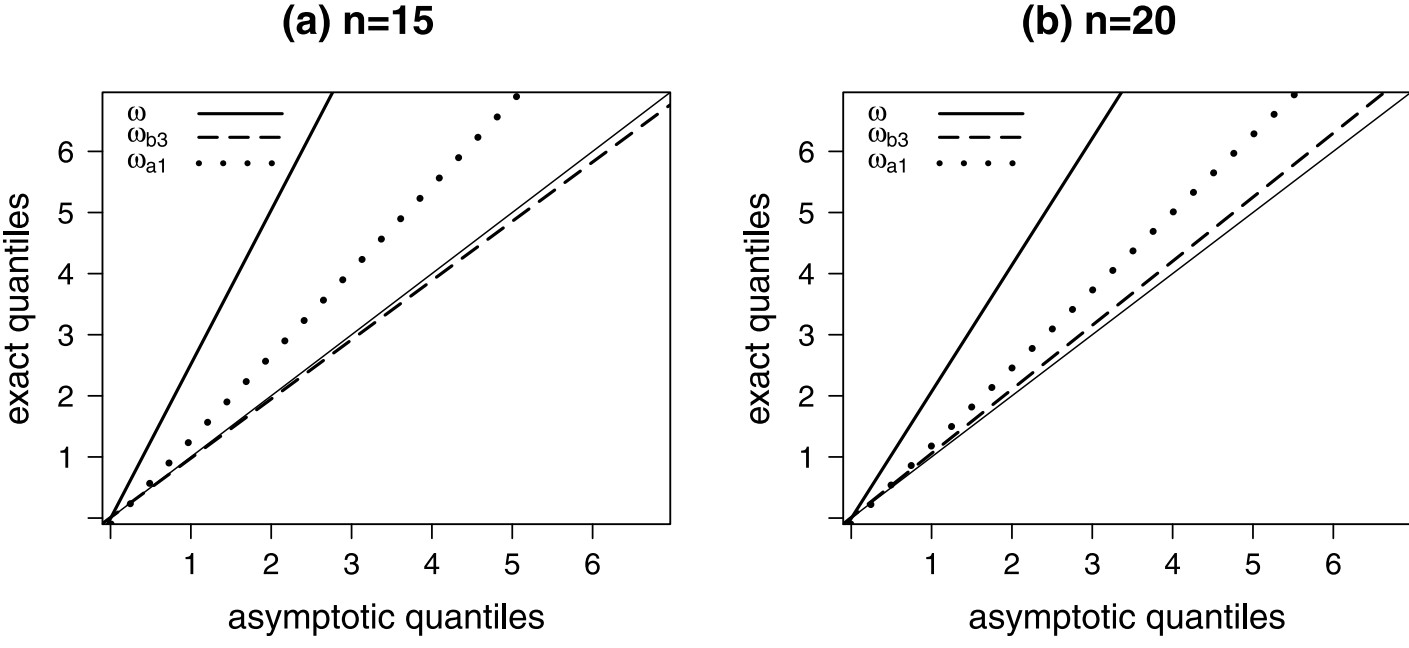

**Fig 1. Quantile-quantile plots, $\mathcal{H}_0 : \beta_4 = 0$.**

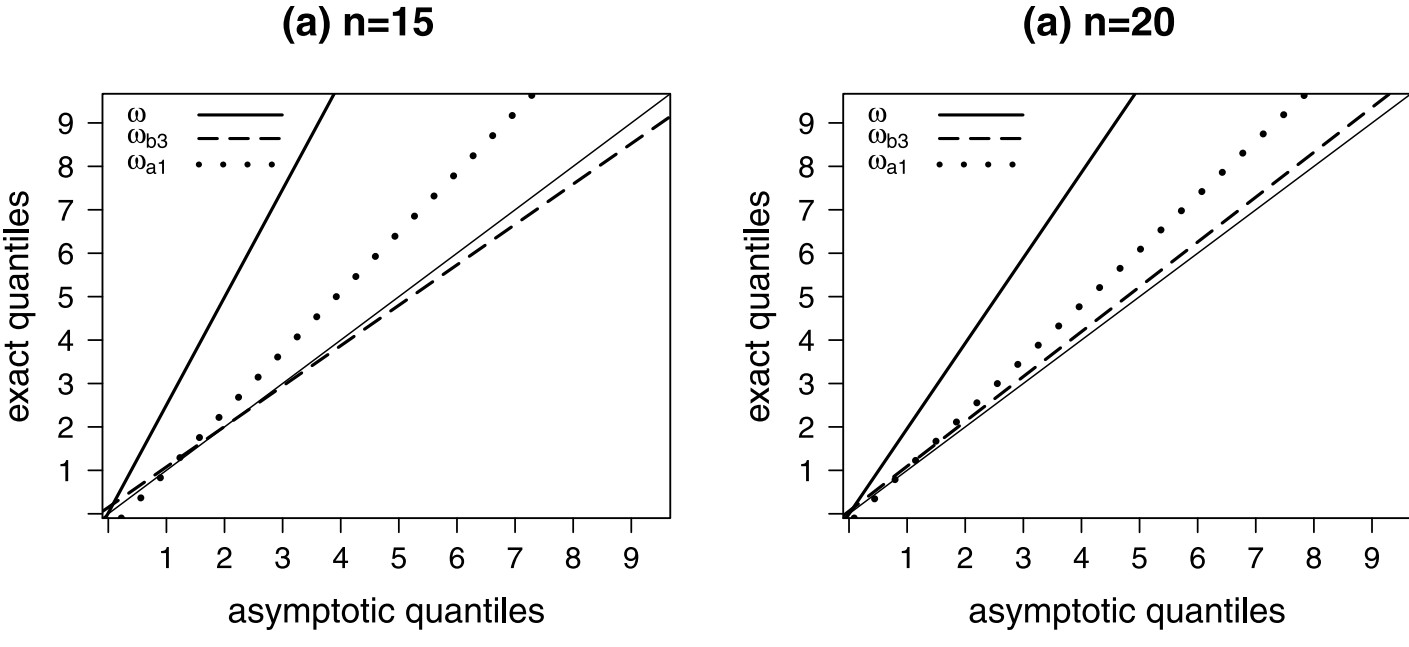

**Fig 2. Quantile-quantile plots, $\mathcal{H}_0: \beta_3 = \beta_4 = 0$.**

('mutiplicative-like'), is very well approximated by the limiting $\chi_1^2$ distribution since the dashed line is very close to the 45° line.

Table 2 contains simulation results for the second scenario, that is, it contains results relative to testing that $\beta_3$ and $\beta_4$ are jointly equal to zero. Here, $l = 2$. Again, the likelihood ratio test is markedly oversized when the sample size is small, even more so than in the previous scenario. For instance, when $\alpha = 10\%$ and $n = 20$, the estimated size of the test equals 31.1%, i.e. the test's empirical size is over three times larger than the nominal significance level. The corrected tests perform much more reliably. Again, overall, the best performing test is that based on our third Bartlett-corrected test statistic ($\omega_{b3}$—'mutiplicative-like'). For instance, when $n = 20$ and $\alpha = 10\%$, its null rejection rate is 10.7%; the corresponding figures for $\omega_{a1}$ and $\omega_{a2}$ (the two alternative corrected tests) are 14.9% and 17.5%, respectively.

Fig 2 contains QQ plots for the second scenario. As in the previous scenario, the null distribution of $\omega$ is poorly approximated by the limiting chi-squared distribution and the approximation works better for $\omega_{b3}$ (the Bartlett-corrected test statistic, 'mutiplicative-like') than for $\omega_{a1}$.

We shall now consider to tests on the coefficients of the precision submodel. The null rejection rates for the third scenario are in Table 3. We test the null hypothesis of fixed precision, i.e. we test $\mathcal{H}_0: \delta_2 = \delta_3 = 0$ which is equivalent to testing $\mathcal{H}_0: \phi_1 = \cdots = \phi_n = \phi$ (the precision parameter is constant across observations). The figures in Table 3 indicate, once again, that testing inferences based on $\omega$ can be quite unreliable when $n$ is small. Overall, the third Bartlett-corrected ('multiplicative-like') test outperforms all other corrected tests. For instance, when $n = 20$ and $\alpha = 10\%$, its null rejection rate is 9.5% whereas those of $\omega_{b1}$ ('ratio-like'), $\omega_{b2}$ ('exponentially adjusted'), $\omega_{a1}$ and $\omega_{a2}$ are 17.1%, 14.4%, 11.7% and 14.1%. We do not present QQ plots for brevity. We note, however, that they show that the null distribution of $\omega_{b3}$ ('mutiplicative-like') is well approximated by the limiting $\chi^2$ distribution.

We also performed simulations using a data generating process that differs from the estimated model, that is, we estimated the tests' non-null rejection rates (powers). We restrict

attention to the likelihood ratio test ($\omega$), the best performing Bartlett-corrected test ($\omega_{b3}$—'mutiplicative-like') and the best performing test obtained using the alternative finite sample correction ($\omega_{a1}$). We consider two sample sizes ($n \in \{20, 40\}$) and two significance levels ($\alpha = 10\%$, 5%). We test $\mathcal{H}_0 : \beta_4 = 0$ (first scenario), but the data are generated using using a value of $\beta_4$ that is different from zero; we denote such a value by $\gamma$. The null hypothesis is thus false. Since some tests are oversized, all testing inferences are carried out using exact (estimated from the size simulations) critical values. The tests' estimated powers for different values of $\gamma$ are presented in Table 4. As expected, the tests become more powerful when the sample size is larger and also as the value of $\gamma$ moves away from zero. Overall, the three tests display similar non-null rejection rates.

We shall now return to the evaluation of the tests' null performances. First, we shall investigate the impact of the number of nuisance parameters on the tests' null behavior. We set the sample size at $n = 40$ and consider the following model:

$$\log\left(\frac{\mu_i}{1 - \mu_i}\right) = \beta_1 + \sum_{j=2}^{p}\beta_j x_{ij}$$
$$\log(\phi_i) = \delta_1 + \delta_2 x_{i2} + \delta_3 x_{i3},$$

$i = 1, \ldots, 40$. We test $\mathcal{H}_0 : \beta_2 = 0$ against $\mathcal{H}_1 : \beta_2 \neq 0$. The covariate $x_2$ is a dummy variable that equals 1 for the first twenty observations and 0 otherwise. The values of all other covariates were obtained as random $\mathcal{U}(-0.5, 0.5)$ draws. Table 5 contains the tests' null rejection rates for $p = 3, 4, 5, 6$. The results show that the likelihood ratio test tends to become progressively more liberal as the number of nuisance parameters increases. In contrast, the corrected tests are much less sensitive to the number of nuisance parameters, especially $\omega_{b3}$ ('mutiplicative-like'), which is the best performing test. Its null rejection rates for the different values of $p$ at the 10% significance level range from 9.9% to 10% whereas those of $\omega$ range between 14.1% ($p = 3$) and 17.1% ($p = 6$).

Second, we shall evaluate the tests' finite sample performances when the null hypothesis includes restrictions on the parameters of both submodels simultaneously. The data generating

**Table 4. Nonnull rejection rates (%), $\mathcal{H}_0 : \beta_4 = 0$.**

| $\gamma$ | $n = 20$ | | | | | | $n = 40$ | | | | | |
|---|---|---|---|---|---|---|---|---|---|---|---|---|
| | $\alpha = 5\%$ | | | $\alpha = 10\%$ | | | $\alpha = 5\%$ | | | $\alpha = 10\%$ | | |
| | $\omega$ | $\omega_{b3}$ | $\omega_{a1}$ | $\omega$ | $\omega_{b3}$ | $\omega_{a1}$ | $\omega$ | $\omega_{b3}$ | $\omega_{a1}$ | $\omega$ | $\omega_{b3}$ | $\omega_{a1}$ |
| -1.5 | 99.1 | 96.9 | 99.1 | 99.7 | 98.4 | 99.8 | 100.0 | 100.0 | 100.0 | 100.0 | 100.0 | 100.0 |
| -1.25 | 95.7 | 94.2 | 95.8 | 98.3 | 97.3 | 98.5 | 99.9 | 99.9 | 99.9 | 100.0 | 100.0 | 100.0 |
| -1.0 | 83.7 | 83.9 | 82.8 | 92.0 | 91.4 | 91.8 | 98.6 | 98.7 | 98.9 | 99.4 | 99.5 | 99.5 |
| -0.75 | 59.3 | 61.9 | 57.8 | 74.1 | 75.7 | 73.8 | 87.8 | 88.8 | 89.3 | 93.6 | 94.0 | 94.3 |
| -0.5 | 31.0 | 32.2 | 29.6 | 45.5 | 47.2 | 44.4 | 54.3 | 56.1 | 56.7 | 68.0 | 68.8 | 69.5 |
| 0.5 | 30.5 | 33.8 | 30.2 | 45.1 | 47.7 | 45.2 | 55.6 | 56.8 | 56.6 | 69.6 | 70.2 | 70.1 |
| 0.75 | 58.7 | 62.9 | 58.1 | 73.6 | 76.0 | 73.2 | 87.7 | 88.6 | 88.5 | 93.8 | 94.0 | 93.9 |
| 1.0 | 83.0 | 85.0 | 82.6 | 91.6 | 92.6 | 92.1 | 98.3 | 98.4 | 98.4 | 99.4 | 99.4 | 99.4 |
| 1.25 | 94.9 | 94.7 | 94.8 | 98.0 | 97.5 | 98.3 | 99.8 | 99.9 | 99.9 | 100.0 | 100.0 | 100.0 |
| 1.5 | 98.8 | 97.6 | 98.8 | 99.6 | 98.8 | 99.7 | 100.0 | 100.0 | 100.0 | 100.0 | 100.0 | 100.0 |

**Table 5. Null rejection rates (%), $\mathcal{H}_0 : \beta_2 = 0$, varying number of nuisance parameters.**

| | $\alpha = 10\%$ | | | | $\alpha = 5\%$ | | | | $\alpha = 1\%$ | | | |
| | $p$ | | | | $p$ | | | | $p$ | | | |
| | 3 | 4 | 5 | 6 | 3 | 4 | 5 | 6 | 3 | 4 | 5 | 6 |
|---|---|---|---|---|---|---|---|---|---|---|---|---|
| $\omega$ | 14.1 | 14.6 | 16.1 | 17.1 | 7.9 | 8.4 | 9.8 | 10.1 | 2.2 | 2.3 | 2.9 | 3.1 |
| $\omega_{b1}$ | 10.6 | 10.9 | 11.5 | 11.9 | 5.2 | 5.8 | 5.9 | 6.4 | 1.0 | 1.2 | 1.3 | 1.3 |
| $\omega_{b2}$ | 10.3 | 10.5 | 11.0 | 11.1 | 5.0 | 5.4 | 5.4 | 5.8 | 0.9 | 1.1 | 1.1 | 1.1 |
| $\omega_{b3}$ | 9.9 | 10.0 | 10.0 | 10.0 | 4.8 | 5.1 | 4.8 | 4.8 | 0.8 | 1.0 | 0.9 | 0.8 |
| $\omega_{a1}$ | 10.1 | 10.3 | 10.4 | 10.6 | 5.0 | 5.3 | 5.3 | 5.4 | 0.9 | 1.1 | 1.1 | 1.1 |
| $\omega_{a2}$ | 10.4 | 10.6 | 10.9 | 11.1 | 5.2 | 5.5 | 5.5 | 5.8 | 0.9 | 1.2 | 1.2 | 1.3 |

process is

$$\begin{aligned}
\log\left(\frac{\mu_i}{1 - \mu_i}\right) &= \beta_1 + \beta_2 x_{i2} + \beta_3 x_{i3} \\
\log(\phi_i) &= \delta_1 + \delta_2 x_{i2} + \delta_3 x_{i3}.
\end{aligned}$$

We consider two different null hypotheses, namely: (i) $\mathcal{H}_0$: $\beta_2 = 0$, $\delta_3 = 0$ ($l = 2$) and (ii) $\mathcal{H}_0$: $\beta_2 = 0$, $\delta_2 = \delta_3 = 0$ ($l = 3$). The corresponding parameter values are (i) $\beta_1 = 1.0$, $\beta_2 = 0$, $\beta_3 = 3.0$, $\delta_1 = 1.7$, $\delta_2 = 0.7$, $\delta_3 = 0$ and (ii) $\beta_1 = 1.5$, $\beta_2 = 0$, $\beta_3 = -1.4$, $\delta_1 = 1.5$, $\delta_2 = \delta_3 = 0$. The covariate values were obtained as random $\mathcal{U}(-0.5, 0.5)$ draws and $n \in \{15, 20, 30, 40\}$. Table 6 contains the tests' null rejection rates. The test based on the Bartlett-corrected test statistic $\omega_{b3}$ ('mutiplicative-like') is the best performer in both cases. For instance, when $l = 2$ and $n = 15$, its null rejection rate at the 10% significance level is 9.5% whereas those of the competing tests range from 11.1% to 25.0%.

**Table 6. Null rejection rates (%), $\mathcal{H}_0 : \beta_2, \delta_3 = 0$ ($l = 2$) and $\mathcal{H}_0 : \beta_2, \delta_2, \delta_3 = 0$ ($l = 3$).**

| | | $\alpha = 10\%$ | | | | $\alpha = 5\%$ | | | | $\alpha = 1\%$ | | | |
| | | $n$ | | | | $n$ | | | | $n$ | | | |
| | | 15 | 20 | 30 | 40 | 15 | 20 | 30 | 40 | 15 | 20 | 30 | 40 |
|---|---|---|---|---|---|---|---|---|---|---|---|---|---|
| $l = 2$ | $\omega$ | 25.0 | 20.4 | 15.3 | 12.4 | 16.4 | 12.6 | 9.0 | 6.7 | 6.8 | 4.5 | 2.4 | 1.6 |
| | $\omega_{b1}$ | 14.6 | 12.6 | 11.3 | 9.8 | 8.2 | 6.9 | 5.8 | 5.0 | 1.9 | 1.2 | 1.2 | 1.0 |
| | $\omega_{b2}$ | 12.8 | 11.5 | 10.9 | 9.6 | 6.9 | 6.1 | 5.6 | 4.8 | 1.3 | 0.9 | 1.1 | 0.9 |
| | $\omega_{b3}$ | 9.5 | 10.2 | 10.4 | 9.5 | 4.3 | 5.0 | 5.2 | 4.7 | 0.6 | 0.6 | 1.0 | 0.9 |
| | $\omega_{a1}$ | 11.1 | 11.6 | 10.8 | 9.5 | 6.0 | 6.2 | 5.5 | 4.8 | 1.2 | 1.1 | 1.2 | 1.0 |
| | $\omega_{a2}$ | 13.1 | 12.2 | 11.0 | 9.6 | 7.2 | 6.6 | 5.7 | 4.9 | 1.5 | 1.3 | 1.2 | 1.0 |
| $l = 3$ | $\omega$ | 23.7 | 18.9 | 13.3 | 12.5 | 15.5 | 11.6 | 7.4 | 6.7 | 5.8 | 3.6 | 1.9 | 1.6 |
| | $\omega_{b1}$ | 13.4 | 11.9 | 9.8 | 9.8 | 7.2 | 6.2 | 5.0 | 4.7 | 1.9 | 1.4 | 1.1 | 1.1 |
| | $\omega_{b2}$ | 11.8 | 11.2 | 9.6 | 9.6 | 6.0 | 5.7 | 4.8 | 4.6 | 1.4 | 1.2 | 1.0 | 1.0 |
| | $\omega_{b3}$ | 9.4 | 10.2 | 9.3 | 9.5 | 4.4 | 5.0 | 4.7 | 4.6 | 0.8 | 1.0 | 0.9 | 1.0 |
| | $\omega_{a1}$ | 7.8 | 8.8 | 8.8 | 9.6 | 3.8 | 4.4 | 4.4 | 4.4 | 0.9 | 0.9 | 0.9 | 0.9 |
| | $\omega_{a2}$ | 9.2 | 9.6 | 9.1 | 9.7 | 4.7 | 4.8 | 4.6 | 4.5 | 1.3 | 1.0 | 0.9 | 1.0 |

**Table 7. Null rejection rates (%), $\mathcal{H}_0 : \beta_2 = \beta_3 = 0$ ($l = 2$); varying correlation between regressors.**

| | | $\alpha = 10\%$ | | | | $\alpha = 5\%$ | | | | $\alpha = 1\%$ | | | |
|---|---|---|---|---|---|---|---|---|---|---|---|---|---|
| | | $n$ | | | | $n$ | | | | $n$ | | | |
| | | 15 | 20 | 30 | 40 | 15 | 20 | 30 | 40 | 15 | 20 | 30 | 40 |
| $\rho = 0.1$ | $\omega$ | 21.5 | 16.3 | 15.2 | 13.1 | 13.7 | 9.6 | 8.5 | 7.2 | 4.7 | 2.9 | 2.2 | 1.6 |
| | $\omega_{b1}$ | 13.5 | 11.2 | 10.8 | 10.5 | 7.2 | 5.8 | 5.2 | 5.2 | 1.8 | 1.4 | 1.1 | 1.0 |
| | $\omega_{b2}$ | 12.2 | 10.6 | 10.4 | 10.3 | 6.4 | 5.5 | 5.1 | 5.1 | 1.4 | 1.2 | 1.0 | 1.0 |
| | $\omega_{b3}$ | 10.6 | 9.9 | 9.9 | 10.2 | 5.1 | 5.2 | 4.8 | 5.0 | 0.9 | 1.1 | 0.9 | 1.0 |
| | $\omega_{a1}$ | 16.9 | 10.4 | 10.3 | 10.5 | 9.8 | 5.7 | 4.9 | 5.3 | 2.9 | 1.2 | 1.0 | 1.0 |
| | $\omega_{a2}$ | 18.1 | 10.7 | 10.6 | 10.6 | 11.0 | 5.9 | 5.1 | 5.4 | 3.6 | 1.4 | 1.1 | 1.0 |
| $\rho = 0.5$ | $\omega$ | 21.7 | 17.1 | 14.8 | 13.3 | 13.4 | 9.8 | 8.2 | 7.2 | 4.2 | 2.6 | 2.1 | 1.6 |
| | $\omega_{b1}$ | 13.3 | 11.4 | 10.6 | 10.6 | 6.8 | 5.7 | 5.3 | 5.5 | 1.2 | 1.2 | 1.2 | 1.1 |
| | $\omega_{b2}$ | 12.1 | 10.8 | 10.3 | 10.4 | 6.1 | 5.3 | 5.1 | 5.4 | 1.0 | 1.1 | 1.1 | 1.0 |
| | $\omega_{b3}$ | 10.5 | 10.1 | 9.8 | 10.3 | 4.8 | 4.9 | 4.8 | 5.3 | 0.8 | 0.9 | 1.0 | 1.0 |
| | $\omega_{a1}$ | 19.2 | 10.5 | 9.8 | 10.5 | 11.8 | 5.3 | 4.9 | 5.5 | 3.6 | 1.1 | 1.1 | 1.1 |
| | $\omega_{a2}$ | 21.3 | 10.9 | 10.1 | 10.6 | 13.6 | 5.5 | 5.1 | 5.5 | 4.9 | 1.2 | 1.1 | 1.1 |
| $\rho = 0.75$ | $\omega$ | 22.3 | 17.0 | 14.7 | 12.8 | 14.0 | 9.8 | 8.3 | 6.8 | 4.4 | 2.6 | 2.3 | 1.8 |
| | $\omega_{b1}$ | 13.3 | 11.2 | 10.7 | 9.9 | 6.9 | 5.8 | 5.5 | 5.0 | 1.3 | 1.3 | 1.0 | 1.2 |
| | $\omega_{b2}$ | 11.9 | 10.7 | 10.3 | 9.8 | 5.9 | 5.4 | 5.3 | 4.9 | 1.0 | 1.2 | 1.0 | 1.1 |
| | $\omega_{b3}$ | 9.7 | 10.1 | 9.9 | 9.7 | 4.4 | 4.9 | 5.0 | 4.9 | 0.6 | 1.1 | 0.9 | 1.1 |
| | $\omega_{a1}$ | 18.4 | 10.2 | 10.0 | 10.9 | 9.9 | 5.3 | 5.2 | 5.1 | 3.3 | 1.3 | 1.0 | 1.1 |
| | $\omega_{a2}$ | 20.5 | 10.6 | 10.2 | 10.0 | 13.4 | 5.6 | 5.3 | 5.1 | 4.8 | 1.4 | 1.0 | 1.1 |
| $\rho = 0.95$ | $\omega$ | 24.0 | 16.8 | 14.4 | 13.1 | 15.3 | 9.3 | 8.2 | 6.9 | 5.3 | 2.6 | 2.2 | 1.5 |
| | $\omega_{b1}$ | 13.8 | 11.0 | 10.4 | 10.5 | 7.1 | 5.4 | 5.3 | 5.1 | 1.7 | 1.2 | 1.2 | 0.9 |
| | $\omega_{b2}$ | 12.0 | 10.5 | 10.1 | 10.3 | 6.1 | 5.1 | 5.1 | 5.1 | 1.2 | 1.1 | 1.2 | 0.9 |
| | $\omega_{b3}$ | 9.4 | 9.8 | 9.8 | 10.1 | 4.3 | 4.7 | 4.8 | 5.0 | 0.8 | 0.9 | 1.0 | 0.9 |
| | $\omega_{a1}$ | 15.5 | 10.3 | 9.8 | 10.5 | 9.3 | 5.0 | 4.9 | 5.2 | 2.6 | 1.1 | 1.2 | 0.9 |
| | $\omega_{a2}$ | 17.9 | 10.7 | 10.0 | 10.9 | 10.6 | 5.2 | 5.1 | 5.3 | 3.7 | 1.1 | 1.3 | 0.9 |

Finally, we shall evaluate the impact of different levels of correlations between regressors on the tests' small sample performance. The model is

$$\log\left(\frac{\mu_i}{1 - \mu_i}\right) = \beta_1 + \beta_2 x_{i2} + \beta_3 x_{i3}$$
$$\log(\phi_i) = \delta_1 + \delta_2 x_{i2}.$$

The values of the two regressors are obtained as random draws from the bivariate normal distribution with mean $(0, 0)^\top$ and covariance matrix $\Sigma$. The diagonal and off-diagonal elements of $\Sigma$ are, respectively, 1 and $\rho$. Hence, $\rho$ is the correlation coefficient between $x_2$ and $x_3$. We test $\mathcal{H}_0$: $\beta_2 = \beta_3 = 0$ ($l = 2$). Data generation was carried out using $\beta_1 = 1.0, \beta_2 = \beta_3 = 0, \delta_1 = 1.7$ and $\delta_2 = 0.1$. Different correlation strengths were considered, ranging from very low to very strong: $\rho \in (0.1, 0.5, 0.75, 0.95)$. The sample sizes are $n \in \{15, 20, 30, 40\}$. Table 7 contains the tests' null rejection rates. Again, the likelihood ratio test $\omega$ is quite liberal when $n$ is small, slightly more so under very strong correlation between the two regressors. The Bartlett-corrected tests perform very well for all correlation values, especially $\omega_{b3}$ ('multiplicative-like'). Its null rejection rates are once again very close to $\alpha$. For instance, when $\rho = 0.75$, $n = 15$ and $\alpha = 10\%$ (5%), the test's null rejection rate is 9.7% (4.4%) whereas that of uncorrected test ($\omega$) is 22.3% (14.0%) and those of alternative tests $\omega_{a1}$ and $\omega_{a2}$ are 18.4% (9.9%) and 20.5% (13.4%),

respectively. It is noteworthy that the null rejection rates of the three Bartlett-corrected tests are insensitive to the level of correlation between regressors. For example, when $n = 15$ and $\alpha = 10\%$, the null rejection rates of $\omega_{b1}$ ('ratio-like'), $\omega_{b2}$ ('exponentially adjusted') and $\omega_{b3}$ ('multiplicative-like') for $\rho = (0.1, 0.5, 0.75, 0.95)$ are in [13.3%, 13.8%], [11.9%, 12.2%] and [9.4%, 10.6%], respectively.

## Behavioral biometrics: Intelligence and atheism

We shall now address the behavioral biometrics issue briefly outlined in the Introduction. The interest lies in modeling the impact of average intelligence on the prevalence of religious disbelievers. General intelligence relates to the ability to reason deductively or inductively, think abstractly, use analogies, synthesize information, and apply it to new domains. It is typically measured by the intelligence quotient (IQ) which is a score obtained from standardized tests. Average IQ scores have been computed for a large number of countries; see e.g. [37, 38]. There is evidence that intelligence negatively correlates with religious belief at the individual level; see e.g. [39]. The negative correlation holds even when religiosity and performance on analytic thinking are measured in separate sessions; see [40]. It also holds when computed from a cross section of nations and from the U.S. states; see [41, 42]. There are evolutionary reasons for the inverse relationship between intelligence and religious belief. For instance, according to the Savanna-IQ Interaction Hypothesis more intelligent individuals are more likely to acquire and espouse evolutionarily novel values and preferences than less intelligent individuals; see [43]. One of such evolutionarily novel values is religious disbelief.

Several regression analysis were performed to measure the net impact of changes in intelligence levels on the prevalence of atheists; see [44] for details. A beta regression analysis was carried out by [13]. They used data on 124 nations and showed that the net impact of average intelligence on the prevalence of religious disbelievers is always positive, gains strength up to a certain level of average intelligence and then weakens. The same data set ($n = 124$) was analyzed by [32] using a beta regression model that includes a parametric mean link function and by [30] using the unit gamma regression model. In what follows, we shall consider a different data set. On page 487 of their paper [13], briefly mention a beta regression analysis that was performed using data on the fifty countries with the largest prevalence of atheists ($n = 50$) which they call 'scenario 3'. Since our interest lies in small sample inferences, we shall pursue that modeling. A novel feature of such data is that they do not include countries for which the prevalence of atheists is very small (close to zero).

The response variable ($y$) is the proportion of atheists in each country and the covariates are: average intelligence quotient ($x_2$), average intelligence quotient squared ($x_3$), life expectancy in 2007 in years ($x_4$), the logarithm of the ratio between trade volume (the sum of imports and exports) and gross national product ($x_5$), and per capita income adjusted for purchasing power parity ($x_6$). Additionally, the following interactions are used: $x_7 = x_5 \times x_6$ and $x_8 = x_4 \times x_5$; the latter was not considered by the original authors. Except for $x_8$, these are the same variables used by [13]. Average intelligence is the independent variable of main interest and the remaining regressors are control variables. Also, $n = 50$ (fifty countries with the largest prevalence of religious disbelievers). The data and computer code used in the empirical analysis that follows can be obtained at https://github.com/acguedes/beta-Bartlett.

[13] fitted the following beta regression model to the data (Model $\mathcal{M}_1$):

$$\log\left(\frac{\mu_i}{1 - \mu_i}\right) = \beta_1 + \beta_2 x_{i2} + \beta_3 x_{i3} + \beta_4 x_{i4} + \beta_5 x_{i5} + \beta_6 x_{i6} + \beta_7 x_{i7}$$

$$\sqrt{\phi_i} = \delta_1 + \delta_2 x_{i2} + \delta_3 x_{i4} + \delta_4 x_{i6}.$$

We noticed that an improved fit according to standard model selection criteria and pseudo-$R^2$ (see below) can be achieved by adding $x_8$ to the mean submodel and by only using $x_2$ in the precision submodel, since the $x_4$ and $x_5$ lose statistical significance when the mean submodel includes the interaction between these two variables. Our model (Model $\mathcal{M}_2$) is then

$$\log\left(\frac{\mu_i}{1-\mu_i}\right) = \beta_1 + \beta_2 x_{i2} + \beta_3 x_{i3} + \beta_4 x_{i4} + \beta_5 x_{i5} + \beta_6 x_{i6} + \beta_7 x_{i7} + \beta_8 x_{i8}$$

$$\sqrt{\phi_i} = \delta_1 + \delta_2 x_{i2}.$$

All parameter estimates of the above model are statistically significant at the 5% significance level according to the $z$ test, and its pseudo-$R^2$, as defined by [1], is superior to that of the model fitted by [13]: 0.3719 vs 0.3216. Model $\mathcal{M}_2$ is also favored by the three most commonly used model selection criteria when compared to Model $\mathcal{M}_1$, AIC (−61.0555 vs −55.3188), AICC (−55.4144 vs −48.3714) and BIC (−41.9352 vs −34.2865). We shall investigate whether $x_4$ and $x_5$ should be excluded from our model by testing whether $\beta_4$ and $\beta_5$ equal zero (individually and jointly). We shall use three tests, namely: the likelihood ratio test ($\omega$), the best performing Bartlett-corrected test ($\omega_{b3}$—'mutiplicative-like') and the best performing test based on the alternative small sample correction ($\omega_{a1}$).

At the outset, we test the exclusion of $x_4$ from Model $\mathcal{M}_2$, that is, we test $\mathcal{H}_0 : \beta_4 = 0$. The $p$-values of the $\omega$, $\omega_{b3}$ and $\omega_{a1}$ tests are 0.0258, 0.0443 and 0.0295, respectively. The first and third tests clearly reject the null hypothesis at $\alpha$ = 5% whereas the $p$-value of the Bartlett-corrected test is very close to 0.05 which renders uncertainty about the exclusion of $x_4$ from the model. Next, we test $\mathcal{H}_0 : \beta_5 = 0$. We obtain the following $p$-values for $\omega$, $\omega_{b3}$ and $\omega_{a1}$: 0.0303, 0.0505 and 0.0332, respectively. The first and third tests clearly reject the removal of $x_5$ from the model at the 5% significance level; the null hypothesis is not rejected by the Bartlett-corrected test. Finally, we test the joint exclusion of both covariates, i.e. we test $\mathcal{H}_0 : \beta_4 = \beta_5 = 0$, and obtain the following $p$-values for $\omega$, $\omega_{b3}$ and $\omega_{a1}$: 0.0726, 0.1121 and 0.0840, respectively. The null hypothesis is not rejected by the three tests at the 5% nominal level, but only the Bartlett-corrected test maintains that inference at the 10% nominal level. That is, such a test provides more evidence in favor of the removal of $x_4$ and $x_5$ from the mean submodel.

Based on the above testing inference, we arrive at the following reduced model (Model $\mathcal{M}_{2R}$), which is our final model:

$$\log\left(\frac{\mu_i}{1-\mu_i}\right) = \beta_1 + \beta_2 x_{i2} + \beta_3 x_{i3} + \beta_4 x_{i6} + \beta_5 x_{i7} + \beta_6 x_{i8}$$

$$\sqrt{\phi_i} = \delta_1 + \delta_2 x_{i2}.$$

The estimates of $\beta_1, \ldots, \beta_6$ (standard errors in parenthesis) are, respectively, 22.9423 (7.6472), −0.7583 (0.1942), 0.0044 (0.0011), 0.1866 (0.0545), −0.0483 (0.0136), 0.0265 (0.0055). For the precision submodel, we obtain $\hat{\delta}_1$ = 22.0333 (5.0312) and $\hat{\delta}_2$ = −0.1934 (0.0498). The model pseudo-$R^2$ is 0.3455; it is higher than that of the model estimated by [13]. Additionally, AIC = −59.8094, AICC = −56.2972 and BIC = −44.5133. It is noteworthy that these criteria clearly favor our reduced model relative to the model presented in [13]; recall that for that model, AIC = −55.3188, AICC = −48.3714 and BIC = −34.2865. The difference in AIC (AICC) [BIC] in favor of Model $\mathcal{M}_{2R}$ is of nearly 5 points (nearly 8 points) [over 10 points]. When the difference in AIC values exceeds 4, one can conclude that there is considerably less support for the model with larger AIC; see [45]. The evidence in favor of our reduced model is thus strong.

Asymptotic confidence intervals with nominal coverage $(1 - \alpha) \times 100\%$ for the parameters of Model $\mathcal{M}_{2R}$ can be obtained using the asymptotic normality of the corresponding

**Table 8. Lower (LLCI) and upper (ULCI) asymptotic confidence intervals limits for the parameters of Model $\mathcal{M}_{2R}$; standard asymptotic confidence interval and confidence intervals constructed using the test statistics $\omega$, $\omega_{b3}$ and $\omega_{a1}$.**

|  | $\text{LLCI}_a$ | $\text{ULCI}_a$ | $\text{LLCI}_\omega$ | $\text{ULCI}_\omega$ | $\text{LLCI}_{\omega_{b3}}$ | $\text{ULCI}_{\omega_{b3}}$ | $\text{LLCI}_{\omega_{a1}}$ | $\text{ULCI}_{\omega_{a1}}$ |
|---|---|---|---|---|---|---|---|---|
| $\beta_1$ | 7.9538 | 37.9302 | 3.7532 | 39.1532 | 3.7532 | 40.3532 | 3.7532 | 41.5532 |
| $\beta_2$ | -1.1389 | -0.3776 | -1.1985 | -0.4794 | -1.1985 | -0.4182 | -1.1985 | -0.4794 |
| $\beta_3$ | 0.0022 | 0.0065 | 0.0011 | 0.0068 | 0.0011 | 0.0069 | 0.0014 | 0.0072 |
| $\beta_4$ | 0.0798 | 0.2935 | 0.0674 | 0.3066 | 0.0622 | 0.3144 | 0.0505 | 0.3066 |
| $\beta_5$ | -0.0749 | -0.0216 | -0.0778 | -0.0184 | -0.0796 | -0.0169 | -0.0781 | -0.0145 |
| $\beta_6$ | 0.0159 | 0.0371 | 0.0137 | 0.0393 | 0.0127 | 0.0403 | 0.0137 | 0.0396 |
| $\delta_1$ | 12.1722 | 31.8941 | 8.7085 | 32.7085 | 8.7085 | 33.5085 | 8.7085 | 32.3085 |
| $\delta_2$ | -0.2910 | -0.0957 | -0.2926 | -0.0566 | -0.2926 | -0.0566 | -0.2926 | -0.0566 |

maximum likelihood estimators. In particular, for $j = 1, \ldots, 6$ and $k = 1, 2$, $\hat{\beta}_j + z_{1-\alpha/2} \, \text{se} \, (\hat{\beta}_j)$ and $\hat{\delta}_k + z_{1-\alpha/2} \, \text{se} \, (\hat{\delta}_k)$ are asymptotic confidence intervals for $\beta_j$ and $\delta_k$ with nominal coverage $(1 - \alpha) \times 100\%$, respectively, the asymptotic standard errors, se, being obtained from Fisher's information matrix inverse evaluated at the maximum likelihood estimates. Here, $z_{1-\alpha/2}$ denotes the $1 - \alpha/2$ standard normal quantile. Table 8 contains the lower and upper limits ($\text{LLCI}_a$ and $\text{ULCI}_a$) of such intervals for the parameters that index Model $\mathcal{M}_{2R}$ for $1 - \alpha = 0.95$. Following [46, Section 3], we also computed approximate confidence intervals based on the test statistics $\omega$, $\omega_{b3}$ and $\omega_{a1}$ which was done by finding the set of parameter values such that the test statistic is smaller than $\chi^2_{1;0.95}$ for each parameter and each test statistic. Such intervals are also presented in Table 8. For instance, the confidence intervals for $\beta_5$ constructed using $\omega_{b3}$ and $\omega_{a1}$ are $[-0.0796, -0.0169]$ and $[-0.0781, -0.0145]$, respectively; the corresponding asymptotic interval estimate is $[-0.0749, -0.0216]$. It is noteworthy that none of the reported confidence intervals contains the value zero.

The model used by [13] (Model $\mathcal{M}_1$) and our reduced model (Model $\mathcal{M}_{2R}$) are non-nested. In order to distinguish between them using a hypothesis test, we performed the *J* test as outlined by [47]. When the test is applied to two non-nested models, say Models $m_1$ and $m_2$, each model is sequentially tested against the other, i.e. we test Model $m_1$ against Model $m_2$, and then we test Model $m_2$ against Model $m_1$. It is thus possible to accept one model as the true model and reject the alternative model, to accept both models (i.e. to conclude that the two models are empirically indistinguishable) or to reject both models. Since the *J* testing inference is reached using the likelihood ratio test, we have also performed the test using the two corrected tests. We first test Model $\mathcal{M}_1$, i.e. the model fitted by [13], against our reduced model (Model $\mathcal{M}_{2R}$). The *p*-values of the tests based on $\omega$, $\omega_{b3}$ and $\omega_{a1}$ are, respectively, 0.0036, 0.0825, 0.0233. All tests reject Model $\mathcal{M}_1$ (i.e. the model used by the authors) at the 10% significance level; the test based on $\omega_{a1}$ ($\omega$) yields rejection at $\alpha = 5\%$ (1%). Next, Model $\mathcal{M}_{2R}$ is tested against Model $\mathcal{M}_1$. The *p*-values of the tests that use $\omega$, $\omega_{b3}$ and $\omega_{a1}$ are 0.0364, 0.1100 and 0.2001, respectively. Interestingly, our model is rejected at the 5% significance level by the likelihood ratio test whereas that inference is reversed when the small sample corrections are applied: the two corrected tests do not yield rejection of the model, nor even at $\alpha = 10\%$. That is, our model is not rejected by the two corrected tests.

It is noteworthy that if we consider the three sets of tests, i.e. the tests of $\mathcal{H}_0 : \beta_4 = 0, \mathcal{H}_0 : \beta_5 = 0$ and $\mathcal{H}_0 : \beta_4 = \beta_5 = 0$, it is clear that the Bartlett-corrected test was the test that most emphatically suggested the removal of both $x_4$ and $x_5$ from the mean submodel of Model $\mathcal{M}_2$.

We constructed a residual normal probability with simulated envelopes using the combined residuals of [48] from our fitted model (Model $\mathcal{M}_{2R}$); see Fig 3. The envelope bands were constructed using 100 replications The plot shows that there is no evidence against the correct specification of our model since all points lie inside the two envelope bands.

In [13], Fig 4, the authors plot an estimate of $\partial\mu_i/\partial x_{i2}$ against a sequence of values of average intelligence by setting all other covariates at their median values. In Fig 4 we present a panel of similar plots each containing two impact curves, namely: (i) that obtained from our reduced model ('new') and (ii) that obtained using the model fitted by [13] ('old'). That is, 'new' and 'old' in Fig 4 refer to Models $\mathcal{M}_{2R}$ and $\mathcal{M}_1$, respectively. Instead of only fixing the covariates other than average intelligence at their median values, we do so at four different quantiles: 0.10, 0.25, 0.50 (median) and 0.75. We note that the two impact curves become more similar (more dissimilar) as the quantile at which the regressors values are set increases (decreases). Such covariates tend to assume larger values for more developed nations since they relate to per capita income, life expectancy and integration to international trade. In particular, the former two variables are highly correlated with economic development. It then follows that one gets a somewhat different functional form of the impact of average intelligence on the prevalence of religious disbelievers in lower income countries when our reduced model is used relative to the model used by the original authors. At the lowest quantile (0.10), the maximal impact computed from our model (Model $\mathcal{M}_{2R}$) is over 11% larger than that obtained using the alternative model (Model $\mathcal{M}_1$). When the covariates values are set at their medians, the figure drops to nearly 4%. To the best of our knowledge, our analysis provides the first measure

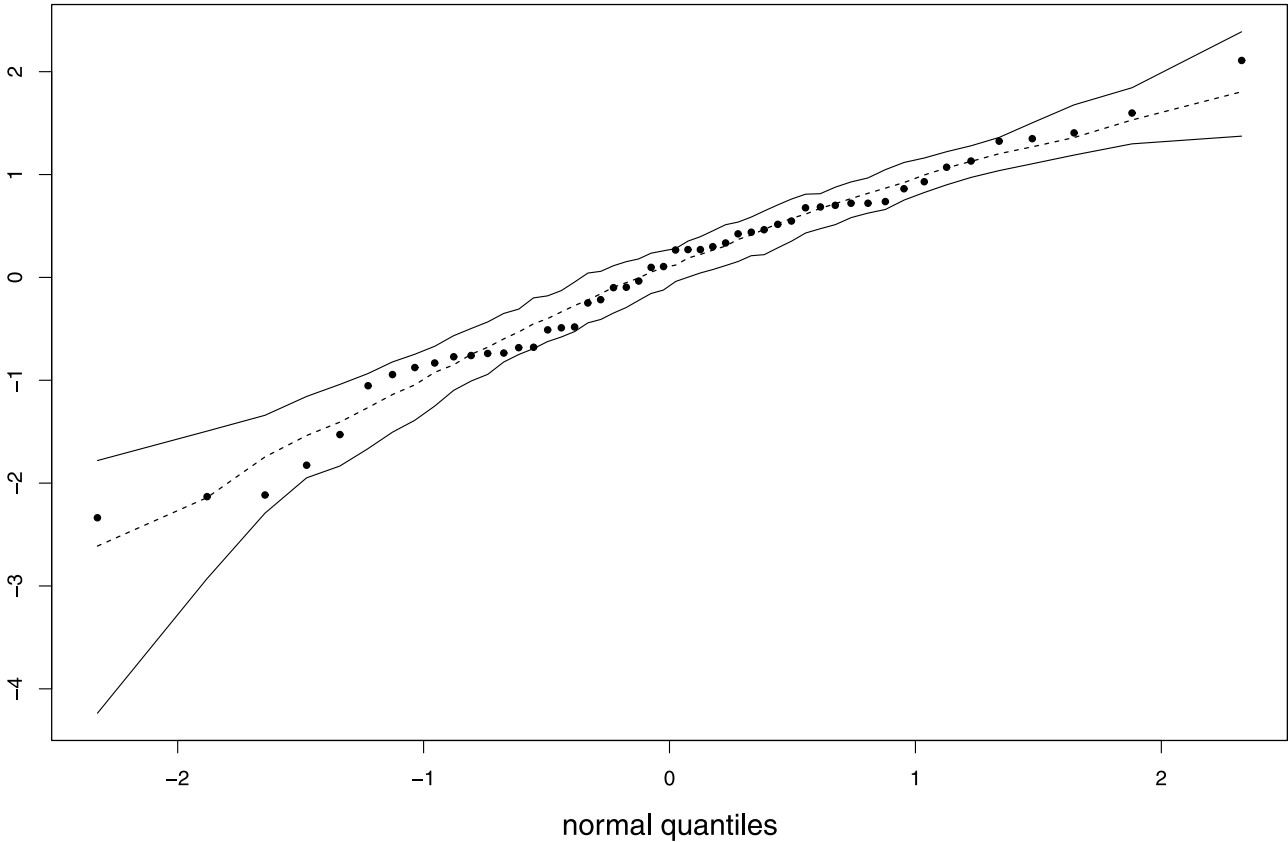

normal quantiles

**Fig 3. Residual normal probability plot.**

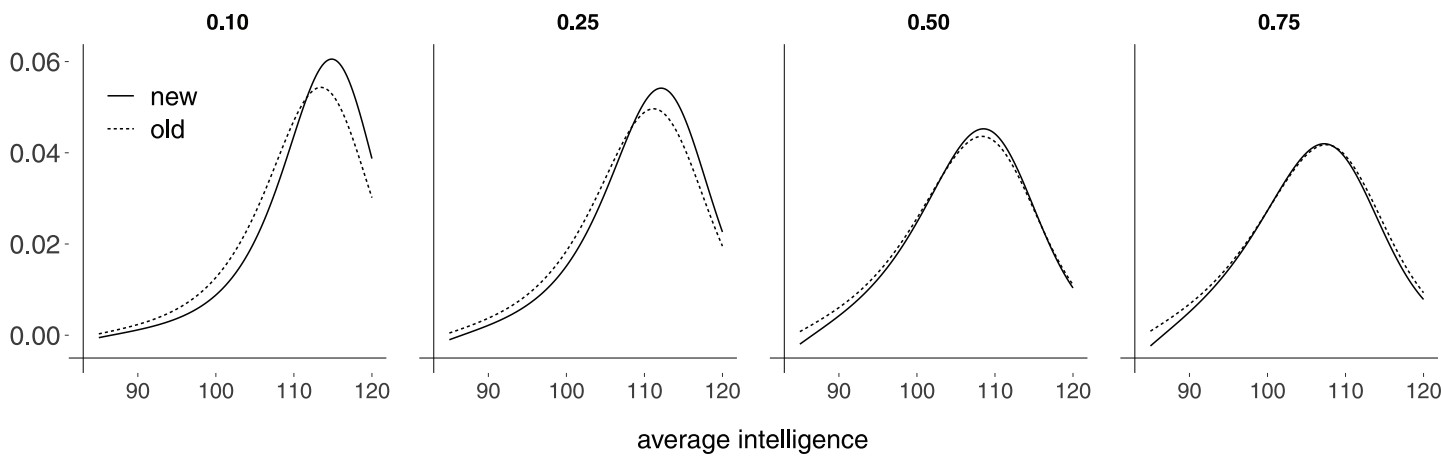

**Fig 4. Impact curves.**

of the decline in the maximal impact of average intelligence on the prevalence of religious disbelievers and also of the changes in the functional form of such an impact as nations become more developed.

## Concluding remarks

The beta regression model is widely used to model responses that assume values in (0, 1). In the initial formulation of the model, the precision parameter was assumed constant for all observations, i.e. all responses in the sample share the same precision. This model became known as the fixed precision beta regression model. A more general and more flexible formulation of the model was later proposed. It allows both distribution parameters to vary across observations. Most empirical applications employ this version of the model, which is known as the varying precision beta regression model. It contains two submodels, one for mean and another for the precision.

In both variants of the regression model, testing inferences are usually performed using the likelihood ratio test. Such a test employs an asymptotic approximation, and as a consequence it can be quite size distorted when the sample size is small. In particular, it tends to be liberal (oversized), i.e. it overrejects the null hypothesis when such a hypothesis is true. Since many applications of the beta regression model are based on samples of small to moderate sizes, it is important to develop alternative tests with superior finite sample behavior, i.e. tests that yield better control of the type I error frequency. [12] derived a Bartlett correction to the likelihood ratio test that can be used to achieve more accurate testing inferences. Their result, nonetheless, only applies to the more restrictive model formulation, namely: the fixed precision model. Since many applications employ the varying precision formulation of the beta regression model, their small sample correction cannot be used. In this paper we derived the Bartlett correction to the likelihood ratio test in full generality. Our correction can thus be used to construct modified likelihood ratio tests to be used in varying precision beta regression analyses. We considered three Bartlett-corrected tests. Monte Carlo simulation evidence revealed that one of such tests typically delivers very accurate inferences even when the sample size is quite small. Its small sample performance was numerically compared to those of two tests that are based on an alternative correction. Overall, the results favor the test that employs the Bartlett correction. A novel feature of our Bartlett-correction tests is that their size distortions are guaranteed to vanish at a faster rate than that of the likelihood ratio test: $O(n^{-2})$ vs $O(n^{-1})$.

We presented and discussed an empirical application that involved an important issue in evolutionary biometrics, namely: the relationship between average intelligence and the prevalence of religious disbelievers. Using data on 50 countries, we showed that by using our Bartlett-corrected testing inferences we arrive at a beta regression model slightly different from that previously used in the literature. It is noteworthy that our model displays superior fit and yields a noticeably different functional form of the impact of intelligence on religious disbelief in low income countries. This empirical application illustrates the usefulness of the Bartlett correction derived in our paper.

A direction for future research is the extension of our analytical results for testing inferences in inflated beta regression models introduced by [49] which include both continuous and discrete components and thus allow for response values that are exactly equal to 0 or 1.

## Appendix: Varying precision beta regression log-likelihood cumulants

We shall now present the varying precision beta regression model log-likelihood cumulants up to fourth order. We shall use lower and upper case case letters to index derivatives of (2) with respect to the components of $\beta$ and $\delta$, respectively. We use tensor notation: $\kappa_{rs} = \mathrm{IE}(\partial^2 \ell(\theta)/\partial\beta_r\partial\beta_s)$, $\kappa_{rst} = \mathrm{IE}(\partial^3 \ell(\theta)/\partial\beta_r\partial\beta_s\partial\beta_t)$, $\kappa_{rstu} = \mathrm{IE}(\partial^4 \ell(\theta)/\partial\beta_r\partial\beta_s\partial\beta_t\partial\beta_u)$, etc., $r, s, t, u = 1, \ldots, k$. Additionally, we use the following notation for derivatives of the above cumulants: $\kappa_{rs}^{(t)} = \partial\kappa_{rs}/\partial\beta_t$, $\kappa_{rs}^{(tu)} = \partial\kappa_{rs}/\partial\beta_t\partial\beta_u$, $\kappa_{rst}^{(u)} = \partial\kappa_{rst}/\partial\beta_u$, etc.

It can be shown that

$$\frac{\partial\mu_i}{\partial\eta_i} = \frac{1}{g_1'(\mu_i)}, \qquad \frac{\partial}{\partial\mu_i}\frac{\partial\mu_i}{\partial\eta_i} = \frac{-g_1''(\mu_i)}{\left(g_1'(\mu_i)\right)^2}, \qquad \frac{\partial}{\partial\mu_i}\left(\frac{\partial\mu_i}{\partial\eta_i}\right)^2 = \frac{-2g_1''(\mu_i)}{\left(g_1'(\mu_i)\right)^3},$$

$$\frac{\partial}{\partial\mu_i}\left(\frac{\partial\mu_i}{\partial\eta_i}\right)^3 = \frac{-3g_1''(\mu_i)}{\left(g_1'(\mu_i)\right)^4}, \qquad \frac{\partial^2}{\partial\mu_i^2}\left(\frac{\partial\mu_i}{\partial\eta_i}\right) = \frac{-g_1'''(\mu_i)g_1'(\mu_i) + 2\left(g_1''(\mu_i)\right)^2}{\left(g_1'(\mu_i)\right)^3},$$

$$\frac{\partial\phi_i}{\partial\zeta_i} = \frac{1}{g_2'(\phi_i)}, \qquad \frac{\partial}{\partial\phi_i}\frac{\partial\phi_i}{\partial\zeta_i} = \frac{-g_2''(\phi_i)}{\left(g_2'(\phi_i)\right)^2}, \qquad \frac{\partial}{\partial\phi_i}\left(\frac{\partial\phi_i}{\partial\zeta_i}\right)^2 = \frac{-2g_2''(\phi_i)}{\left(g_2'(\phi_i)\right)^3},$$

$$\frac{\partial}{\partial\phi_i}\left(\frac{\partial\phi_i}{\partial\zeta_i}\right)^3 = \frac{-3g_2''(\phi_i)}{\left(g_2'(\phi_i)\right)^4}, \qquad \frac{\partial^2}{\partial\phi_i^2}\left(\frac{\partial\phi_i}{\partial\zeta_i}\right) = \frac{-g_2'''(\phi_i)g_2'(\phi_i) + 2\left(g_2''(\phi_i)\right)^2}{\left(g_2'(\phi_i)\right)^3}.$$

Let $w_i = \psi'(\mu_i\phi_i) + \psi'((1 - \mu_i)\phi_i)$, $m_i = \psi''(\mu_i\phi_i) - \psi''((1 - \mu_i)\phi_i)$ and also

$$a_i = 3\left(\frac{\partial}{\partial\mu_i}\frac{\partial\mu_i}{\partial\eta_i}\right)\left(\frac{\partial\mu_i}{\partial\eta_i}\right)^2, \qquad t_i = 3\left(\frac{\partial}{\partial\phi_i}\frac{\partial\phi_i}{\partial\zeta_i}\right)\left(\frac{\partial\phi_i}{\partial\zeta_i}\right)^2,$$

$$c_i = \phi_i[\mu_i w_i - \psi'((1 - \mu_i)\phi_i)], \quad b_i = \frac{\partial\mu_i}{\partial\eta_i}\left[\left(\frac{\partial^2}{\partial\mu_i^2}\frac{\partial\mu_i}{\partial\eta_i}\right)\frac{\partial\mu_i}{\partial\eta_i} + \left(\frac{\partial}{\partial\mu_i}\frac{\partial\mu_i}{\partial\eta_i}\right)^2\right],$$

$$v_i = \frac{\partial\phi_i}{\partial\zeta_i}\left[\left(\frac{\partial^2}{\partial\phi_i^2}\frac{\partial\phi_i}{\partial\zeta_i}\right)\frac{\partial\phi_i}{\partial\zeta_i} + \left(\frac{\partial}{\partial\phi_i}\frac{\partial\phi_i}{\partial\zeta_i}\right)^2\right],$$

$$d_i = (1 - \mu_i)^2\psi'((1 - \mu_i)\phi_i) + \mu^2\psi'(\mu_i\phi_i) - \psi'(\phi_i), \quad s_i = (1 - \mu_i)^3\psi''((1 - \mu_i)\phi_i)$$

$$+ \mu_i^3\psi''(\mu_i\phi_i) - \psi''(\phi_i), \quad u_i = -\phi_i\left[2w_i + \phi\frac{\partial w_i}{\partial\phi_i}\right], \quad r_i = \left[2\frac{\partial\mu_i^*}{\partial\phi_i} + \phi\frac{\partial^2\mu_i^*}{\partial\phi_i^2}\right]\frac{\partial\mu_i}{\partial\eta_i},$$

where $\psi'(\cdot)$ and $\psi''(\cdot)$ is the trigamma and tetragamma functions, respectively. The following

derivatives are needed for obtaining the log-likelihood cumulants:

$$\frac{\partial \mu_i^*}{\partial \mu_i} = \phi_i \psi'(\mu_i \phi_i) + \phi_i \psi'((1-\mu_i)\phi_i) = \phi_i w_i, \quad \frac{\partial \mu_i^*}{\partial \phi_i} = \mu_i \psi'(\mu_i \phi_i) - (1-\mu_i)\psi'((1-\mu_i)\phi_i)$$

$$= \frac{c_i}{\phi_i}, \quad \frac{\partial \mu_i^\dagger}{\partial \mu_i} = -\phi_i \psi'((1-\mu_i)\phi_i), \quad \frac{\partial \mu_i^\dagger}{\partial \phi_i} = (1-\mu_i)\psi'((1-\mu_i)\phi_i) - \psi'(\phi_i),$$

$$\frac{\partial w_i}{\partial \mu_i} = \phi_i \psi''(\mu_i \phi_i) - \phi_i \psi''((1-\mu_i)\phi_i) = \phi_i m_i,$$

$$\frac{\partial w_i}{\partial \phi_i} = \mu_i \psi''(\mu_i \phi_i) + (1-\mu_i)\psi''((1-\mu_i)\phi_i),$$

$$\frac{\partial m_i}{\partial \mu_i} = \phi_i \psi'''(\mu_i \phi_i) + \phi_i \psi'''((1-\mu_i)\phi_i), \quad \frac{\partial m_i}{\partial \phi_i} = \mu_i \psi'''(\mu_i \phi_i) - (1-\mu_i)\psi'''((1-\mu_i)\phi_i),$$

$$\frac{\partial a_i}{\partial \mu_i} = 3\left(\frac{\partial \mu_i}{\partial \eta_i}\right)\left[\left(\frac{\partial^2 \mu_i}{\partial \mu_i^2}\frac{\partial \mu_i}{\partial \eta_i}\right)\left(\frac{\partial \mu_i}{\partial \eta_i}\right) + 2\left(\frac{\partial}{\partial \mu_i}\frac{\partial \mu_i}{\partial \eta_i}\right)^2\right], \quad \frac{\partial u_i}{\partial \mu_i} = -\phi_i^2\left(3m_i + \phi_i\frac{\partial m_i}{\partial \phi_i}\right),$$

$$\frac{\partial t_i}{\partial \phi_i} = 3\left(\frac{\partial \phi_i}{\partial \zeta_i}\right)\left[\left(\frac{\partial^2 \phi_i}{\partial \phi_i^2}\frac{\partial \phi_i}{\partial \zeta_i}\right)\left(\frac{\partial \phi_i}{\partial \zeta_i}\right) + 2\left(\frac{\partial}{\partial \phi_i}\frac{\partial \phi_i}{\partial \zeta_i}\right)^2\right], \quad \frac{\partial}{\partial \mu_i}\frac{\partial w_i}{\partial \phi_i} = m_i + \phi_i\frac{\partial m_i}{\partial \phi_i},$$

$$\frac{\partial b_i}{\partial \mu_i} = \left(\frac{\partial}{\partial \mu_i}\frac{\partial \mu_i}{\partial \eta_i}\right)^3 + \left(\frac{\partial \mu_i}{\partial \eta_i}\right)\left[\left(\frac{\partial^3 \mu_i}{\partial \mu_i^3}\frac{\partial \mu_i}{\partial \eta_i}\right)\left(\frac{\partial \mu_i}{\partial \eta_i}\right) + 4\left(\frac{\partial^2 \mu_i}{\partial \mu_i^2}\frac{\partial \mu_i}{\partial \eta_i}\right)\left(\frac{\partial}{\partial \mu_i}\frac{\partial \mu_i}{\partial \eta_i}\right)\right],$$

$$\frac{\partial v_i}{\partial \phi_i} = \left(\frac{\partial}{\partial \phi_i}\frac{\partial \phi_i}{\partial \zeta_i}\right)^3 + \left(\frac{\partial \phi_i}{\partial \zeta_i}\right)\left[\left(\frac{\partial^3 \phi_i}{\partial \phi_i^3}\frac{\partial \phi_i}{\partial \zeta_i}\right)\left(\frac{\partial \phi_i}{\partial \zeta_i}\right) + 4\left(\frac{\partial^2 \phi_i}{\partial \phi_i^2}\frac{\partial \phi_i}{\partial \zeta_i}\right)\left(\frac{\partial}{\partial \phi_i}\frac{\partial \phi_i}{\partial \zeta_i}\right)\right],$$

$$\frac{\partial^2 \mu_i^*}{\partial \phi_i^2} = \mu_i^2 \psi''(\mu_i \phi_i) - (1-\mu_i)^2 \psi''((1-\mu_i)\phi_i), \quad \frac{\partial c_i}{\partial \mu_i} = \phi_i\left(w_i + \phi_i\frac{\partial w_i}{\partial \phi_i}\right),$$

$$\frac{\partial^2 w_i}{\partial \phi_i^2} = \mu_i^2 \psi'''(\mu_i \phi_i) + (1-\mu_i)^2 \psi'''((1-\mu_i)\phi_i), \quad \frac{\partial c_i}{\partial \phi_i} = \frac{\partial \mu_i^*}{\partial \phi_i} + \phi_i\frac{\partial^2 \mu_i^*}{\partial \phi_i^2},$$

$$\frac{\partial^3 \mu_i^*}{\partial \phi_i^3} = \mu_i^3 \psi'''(\mu_i \phi_i) - (1-\mu_i)^3 \psi'''((1-\mu_i)\phi_i), \quad \frac{\partial s_i}{\partial \mu_i} = 3\frac{\partial^2 \mu_i^*}{\partial \phi_i^2} + \phi_i\frac{\partial^3 \mu_i^*}{\partial \phi_i^3},$$

$$\frac{\partial s_i}{\partial \phi_i} = \mu_i^4 \psi'''(\mu_i \phi_i) + (1-\mu_i)^4 \psi'''((1-\mu_i)\phi_i) - \psi'''(\phi_i),$$

$$\frac{\partial r_i}{\partial \mu_i} = \left(2\frac{\partial \mu_i^*}{\partial \phi_i} + \phi_i\frac{\partial^2 \mu_i^*}{\partial \phi_i^2}\right)\left(\frac{\partial}{\partial \mu_i}\frac{\partial \mu_i}{\partial \eta_i}\right) + \left(2w_i + 4\frac{\partial w_i}{\partial \phi_i} + \phi_i^2\frac{\partial^2 w_i}{\partial \phi_i^2}\right)\left(\frac{\partial \mu_i}{\partial \eta_i}\right),$$

$$\frac{\partial r_i}{\partial \phi_i} = \left(3\frac{\partial^2 \mu_i^*}{\partial \phi_i^2} + \phi_i\frac{\partial^3 \mu_i^*}{\partial \phi_i^3}\right)\left(\frac{\partial \mu_i}{\partial \eta_i}\right), \quad \frac{\partial u_i}{\partial \phi_i} = -2w_i - \phi_i\left(4\frac{\partial w_i}{\partial \phi_i} + \phi_i\frac{\partial^2 w_i}{\partial \phi_i^2}\right).$$

The log-likelihood derivatives with respect to the components of $\theta = (\beta^\top, \delta^\top)^\top$ are given by

$$U_{rs} = \sum_{i=1}^{n}\left\{\left[-\phi_i^2 w_i\left(\frac{\partial\mu_i}{\partial\eta_i}\right) - \phi_i(y_i^* - \mu_i^*)\left(\frac{\partial}{\partial\mu_i}\frac{\partial\mu_i}{\partial\eta_i}\right)\right]\left(\frac{\partial\mu_i}{\partial\eta_i}\right)\right\}x_{ir}x_{is},$$

$$U_{rR} = \sum_{i=1}^{n}[-c_i + (y_i^* - \mu_i^*)]\left(\frac{\partial\phi_i}{\partial\zeta_i}\right)\left(\frac{\partial\mu_i}{\partial\eta_i}\right)x_{ir}z_{iR},$$

$$U_{RS} = \sum_{i=1}^{n}\left\{-d_i\left(\frac{\partial\phi_i}{\partial\zeta_i}\right)^2 + [(y_i^\dagger - \mu_i^\dagger) + \mu_i(y_i^* - \mu_i^*)]\left(\frac{\partial}{\partial\phi_i}\frac{\partial\phi_i}{\partial\zeta_i}\right)\left(\frac{\partial\phi_i}{\partial\zeta_i}\right)\right\}z_{iR}z_{iS},$$

$$U_{rst} = \sum_{i=1}^{n}\left\{-\phi_i\left[\phi_i w_i a_i + \phi_i^2 m_i\left(\frac{\partial\mu_i}{\partial\eta_i}\right)^3 + (y_i^* - \mu_i^*)b_i\right]\right\}x_{ir}x_{is}x_{it},$$

$$U_{rsR} = \sum_{i=1}^{n}\left\{\left[-c_i\left(\frac{\partial}{\partial\eta_i}\frac{\partial\mu_i}{\partial\eta_i}\right) + (y_i^* - \mu_i^*)\left(\frac{\partial}{\partial\eta_i}\frac{\partial\mu_i}{\partial\eta_i}\right) + u_i\left(\frac{\partial\mu_i}{\partial\eta_i}\right)\right]\left(\frac{\partial\mu_i}{\partial\eta_i}\right)\left(\frac{\partial\phi_i}{\partial\zeta_i}\right)\right\}x_{ir}$$
$$\times x_{is}z_{iR},$$

$$U_{rRS} = \sum_{i=1}^{n}\left\{\left(\frac{\partial}{\partial\zeta_i}\frac{\partial\phi_i}{\partial\zeta_i}\right)\left(\frac{\partial\phi_i}{\partial\zeta_i}\right)\left(\frac{\partial\mu_i}{\partial\eta_i}\right)[(y_i^* - \mu_i^*) - c_i] - \left(\frac{\partial\phi_i}{\partial\zeta_i}\right)^2 r_i\right\}x_{ir}z_{iR}$$
$$\times z_{iS},$$

$$U_{RST} = \sum_{i=1}^{n}\left\{-s_i\left(\frac{\partial\phi_i}{\partial\zeta_i}\right)^3 - d_i\left(\frac{\partial}{\partial\phi_i}\left(\frac{\partial\phi_i}{\partial\zeta_i}\right)^2\right)\left(\frac{\partial\phi_i}{\partial\zeta_i}\right) - d_i\left(\frac{\partial}{\partial\phi_i}\frac{\partial\phi_i}{\partial\zeta_i}\right)\left(\frac{\partial\phi_i}{\partial\zeta_i}\right)^2\right.$$
$$\left.+[(y_i^\dagger - \mu_i^\dagger) + \mu_i(y_i^* - \mu_i^*)]v_i\right\}$$
$$\times z_{iR}z_{iS}z_{iT},$$

$$U_{rstu} = \sum_{i=1}^{n}\left\{-\phi_i\left[\phi_i^2\left(m_i\frac{\partial}{\partial\mu_i}\left(\frac{\partial\mu_i}{\partial\eta_i}\right)^3 + \frac{\partial m_i}{\partial\mu_i}\left(\frac{\partial\mu_i}{\partial\eta_i}\right)^3\right) + \phi_i\left[\left(\frac{\partial a_i}{\partial\mu_i} + b_i\right)w_i\right.\right.\right.$$
$$\left.\left.\left.+\frac{\partial w_i}{\partial\mu_i}a_i\right] - (y_i^* - \mu_i^*)\frac{\partial b_i}{\partial\mu_i}\right]\frac{\partial\mu_i}{\partial\eta_i}\right\}x_{ir}x_{is}x_{it}x_{iu},$$

$$U_{rstR} = \sum_{i=1}^{n}\left\{\left[-\phi_i\left[\phi_i\left(3m_i + \phi_i\left(\frac{\partial m_i}{\partial\phi_i}\right)\right)\left(\frac{\partial\mu_i}{\partial\eta_i}\right)^3 + a_i\left(2w_i + \phi_i\left(\frac{\partial w_i}{\partial\phi_i}\right)\right)\right.\right.\right.$$
$$\left.\left.\left.+b_i\left(\frac{\partial\mu_i^*}{\partial\phi_i}\right)\right] + (y_i^* - \mu_i^*)b_i\right]\left(\frac{\partial\phi_i}{\partial\zeta_i}\right)\right\}x_{ir}x_{is}x_{it}z_{iR},$$

$$U_{rsRS} = \sum_{i=1}^{n}\left\{\left[[(y_i^* - \mu_i^*) - c_i]\left(\frac{\partial}{\partial\mu_i}\frac{\partial\mu_i}{\partial\eta_i}\right)\left(\frac{\partial}{\partial\phi_i}\frac{\partial\phi_i}{\partial\zeta_i}\right)\left(\frac{\partial\phi_i}{\partial\zeta_i}\right) - \left(\frac{\partial r_i}{\partial\mu_i}\right)\left(\frac{\partial\phi_i}{\partial\zeta_i}\right)^2\right.\right.$$
$$\left.\left.-\left(\frac{\partial\mu_i^*}{\partial\phi_i}\right)\left(\frac{\partial\mu_i}{\partial\eta_i}\right)\left(\frac{\partial\phi_i}{\partial\zeta_i}\right)^2 + [u_i + (y_i^* - \mu_i^*)]\left(\frac{\partial\mu_i}{\partial\eta_i}\right)\left(\frac{\partial\phi_i}{\partial\zeta_i}\right)\left(\frac{\partial}{\partial\phi_i}\frac{\partial\phi_i}{\partial\zeta_i}\right)\right]\left(\frac{\partial\mu_i}{\partial\eta_i}\right)\right\}$$
$$\times x_{ir}x_{is}z_{iR}z_{iS},$$

$$U_{rRST} = \sum_{i=1}^{n}\left\{[(y_i^* - \mu_i^*) - c_i]\left(\frac{\partial\mu_i}{\partial\eta_i}\right)v_i - r_i\left(\frac{\partial}{\partial\phi_i}\left(\frac{\partial\phi_i}{\partial\zeta_i}\right)^3\right) - \left(\frac{\partial s_i}{\partial\mu_i}\right)\left(\frac{\partial\mu_i}{\partial\eta_i}\right)\left(\frac{\partial\phi_i}{\partial\zeta_i}\right)^3\right\}$$
$$\times x_{ir}z_{iR}z_{iS}z_{iT}, \quad U_{RSTU} = \sum_{i=1}^{n}\left\{\left[-\left(\frac{\partial s_i}{\partial\phi_i}\right)\left(\frac{\partial\phi_i}{\partial\zeta_i}\right)^3 - 2s_i\left(\frac{\partial}{\partial\phi_i}\left(\frac{\partial\phi_i}{\partial\zeta_i}\right)^3\right)\right.\right.$$
$$\left.\left.-d_i\left(\left(\frac{\partial t_i}{\partial\phi_i}\right) + v_i\right) + [(y_i^\dagger - \mu_i^\dagger) + \mu_i(y_i^* - \mu_i^*)]\left(\frac{\partial v_i}{\partial\phi_i}\right)\right]\left(\frac{\partial\phi_i}{\partial\zeta_i}\right)\right\}z_{iR}z_{iS}z_{iT}z_{iU}.$$

Using the above results, we arrive, after long derivations, at the following expressions for the relevant varying precision beta regression model cumulants:

$$\kappa_{rs} = -\sum_{i=1}^{n}\left\{\phi_i^2 w_i\left(\frac{\partial\mu_i}{\partial\eta_i}\right)^2\right\}x_{ir}x_{is}, \quad \kappa_{rR} = -\sum_{i=1}^{n}\left\{c_i\left(\frac{\partial\mu_i}{\partial\eta_i}\right)\left(\frac{\partial\phi_i}{\partial\zeta_i}\right)\right\}x_{ir}z_{iR},$$

$$\kappa_{RS} = -\sum_{i=1}^{n}\left\{d_i\left(\frac{\partial\phi_i}{\partial\zeta_i}\right)^2\right\}z_{iR}z_{iS}, \quad \kappa_{rst} = -\sum_{i=1}^{n}\left\{\phi_i^3 m_i\left(\frac{\partial\mu_i}{\partial\eta_i}\right)^3 + \phi_i^2 w_i a_i\right\}x_{ir}x_{is}x_{it},$$

$$\kappa_{rsR} = \sum_{i=1}^{n}\left\{\left[u_i\left(\frac{\partial\mu_i}{\partial\eta_i}\right) - c_i\left(\frac{\partial}{\partial\mu_i}\frac{\partial\mu_i}{\partial\eta_i}\right)\right]\left(\frac{\partial\mu_i}{\partial\eta_i}\right)\left(\frac{\partial\phi_i}{\partial\zeta_i}\right)\right\}x_{ir}x_{is}z_{iR},$$

$$\kappa_{rRS} = -\sum_{i=1}^{n}\left\{r_i\left(\frac{\partial\phi_i}{\partial\zeta_i}\right)^2 + c_i\left(\frac{\partial}{\partial\phi_i}\frac{\partial\phi_i}{\partial\zeta_i}\right)\left(\frac{\partial\mu_i}{\partial\eta_i}\right)\left(\frac{\partial\phi_i}{\partial\zeta_i}\right)\right\}x_{ir}z_{iR}z_{iS},$$

$$\kappa_{RST} = -\sum_{i=1}^{n}\left\{s_i\left(\frac{\partial\phi_i}{\partial\zeta_i}\right)^3 + d_i\frac{\partial}{\partial\phi_i}\left(\frac{\partial\phi_i}{\partial\zeta_i}\right)^3\right\}z_{iR}z_{iS}z_{iT}, \quad \kappa_{rstu} = \sum_{i=1}^{n}\left\{-\phi_i\left[\phi_i^2\left(\frac{\partial m_i}{\partial\mu_i}\right)\right.\right.$$

$$\times\left(\frac{\partial\mu_i}{\partial\eta_i}\right)^3 + m_i\frac{\partial}{\partial\mu_i}\left(\frac{\partial\mu_i}{\partial\eta_i}\right)^3\right) + \phi_i\left(\left(\frac{\partial a_i}{\partial\mu_i} + b_i\right)w_i + \frac{\partial w_i}{\partial\mu_i}a_i\right)\left]\frac{\partial\mu_i}{\partial\eta_i}\right\}x_{ir}x_{is}x_{it}x_{iu},$$

$$\kappa_{rstR} = \sum_{i=1}^{n}\left\{-\phi_i\left[\phi_i\left(3m_i + \phi_i\left(\frac{\partial m_i}{\partial\phi_i}\right)\right)\left(\frac{\partial\mu_i}{\partial\eta_i}\right)^3 + a_i\left(2w_i + \phi_i\left(\frac{\partial w_i}{\partial\phi_i}\right)\right) + b_i\left(\frac{\partial\mu_i^*}{\partial\phi_i}\right)\right]\right.$$

$$\times\left(\frac{\partial\phi_i}{\partial\zeta_i}\right)\right\}x_{ir}x_{is}x_{it}z_{iR}, \quad \kappa_{rsRS} = \sum_{i=1}^{n}\left\{-c_i\left(\frac{\partial}{\partial\mu_i}\frac{\partial\mu_i}{\partial\eta_i}\right)\left(\frac{\partial}{\partial\phi_i}\frac{\partial\phi_i}{\partial\zeta_i}\right)\left(\frac{\partial\phi_i}{\partial\zeta_i}\right)\right.$$

$$-\left(\frac{\partial r_i}{\partial\mu_i}\right)\left(\frac{\partial\phi_i}{\partial\zeta_i}\right)^2 - \left(\frac{\partial\mu_i^*}{\partial\phi_i}\right)\left(\frac{\partial\mu_i}{\partial\eta_i}\right)\left(\frac{\partial\phi_i}{\partial\zeta_i}\right)^2 + u_i\left(\frac{\partial\mu_i}{\partial\eta_i}\right)\left(\frac{\partial\phi_i}{\partial\zeta_i}\right)\left(\frac{\partial}{\partial\phi_i}\frac{\partial\phi_i}{\partial\zeta_i}\right)\right\}\frac{\partial\mu_i}{\partial\eta_i}$$

$$\times x_{ir}x_{is}z_{iR}z_{iS}, \quad \kappa_{rRST} = -\sum_{i=1}^{n}\left\{c_i\left(\frac{\partial\mu_i}{\partial\eta_i}\right)v_i + r_i\left(\frac{\partial}{\partial\phi_i}\left(\frac{\partial\phi_i}{\partial\zeta_i}\right)^3\right) + \left(\frac{\partial s_i}{\partial\mu_i}\right)\left(\frac{\partial\mu_i}{\partial\eta_i}\right)\right.$$

$$\times\left(\frac{\partial\phi_i}{\partial\zeta_i}\right)^3\right\}x_{ir}z_{iR}z_{iS}z_{iT}, \quad \kappa_{RSTU} = -\sum_{i=1}^{n}\left\{\left[\left(\frac{\partial s_i}{\partial\phi_i}\right)\left(\frac{\partial\phi_i}{\partial\zeta_i}\right)^3 + 2s_i\left(\frac{\partial}{\partial\phi_i}\left(\frac{\partial\phi_i}{\partial\zeta_i}\right)^3\right)\right.\right.$$

$$+d_i\left(\frac{\partial t_i}{\partial\phi_i} + v_i\right)\left]\left(\frac{\partial\phi_i}{\partial\zeta_i}\right)\right\}z_{iR}z_{iS}z_{iT}z_{iU}.$$

Also, we obtained the following expressions for the first order derivatives of the log-likelihood cumulants:

$$\kappa_{rs}^{(u)} = \sum_{i=1}^{n}\left\{-\phi_i^2\left[\phi_i m_i\left(\frac{\partial\mu_i}{\partial\eta_i}\right)^3 + \frac{2}{3}w_i a_i\right]\right\}x_{ir}x_{is}x_{iu}, \quad \kappa_{rs}^{(R)} = \sum_{i=1}^{n}\left\{u_i\left(\frac{\partial\mu_i}{\partial\eta_i}\right)^2\left(\frac{\partial\phi_i}{\partial\zeta_i}\right)\right\}$$
$$\times x_{ir}x_{is}z_{iR},$$

$$\kappa_{rR}^{(u)} = -\sum_{i=1}^{n}\left\{\left[\left(\frac{\partial c_i}{\partial\mu_i}\right)\left(\frac{\partial\mu_i}{\partial\eta_i}\right) + c_i\left(\frac{\partial}{\partial\mu_i}\frac{\partial\mu_i}{\partial\eta_i}\right)\right]\left(\frac{\partial\mu_i}{\partial\eta_i}\right)\left(\frac{\partial\phi_i}{\partial\zeta_i}\right)\right\}x_{ir}x_{iu}z_{iR},$$

$$\kappa_{rR}^{(S)} = -\sum_{i=1}^{n}\left\{\left[\left(\frac{\partial c_i}{\partial\phi_i}\right)\left(\frac{\partial\phi_i}{\partial\zeta_i}\right) + c_i\left(\frac{\partial}{\partial\phi_i}\frac{\partial\phi_i}{\partial\zeta_i}\right)\right]\left(\frac{\partial\mu_i}{\partial\eta_i}\right)\left(\frac{\partial\phi_i}{\partial\zeta_i}\right)\right\}x_{ir}z_{iR}z_{iS},$$

$$\kappa_{RS}^{(u)} = -\sum_{i=1}^{n}\left\{r_i\left(\frac{\partial\phi_i}{\partial\zeta_i}\right)^2\right\}x_{iu}z_{iR}z_{iS}, \quad \kappa_{RS}^{(T)} = -\sum_{i=1}^{n}\left\{s_i\left(\frac{\partial\phi_i}{\partial\zeta_i}\right)^3 + \frac{2}{3}d_i t_i\right\}z_{iR}z_{iS}z_{iT},$$

$$\kappa_{rst}^{(u)} = \sum_{i=1}^{n}\left\{-\phi_i^2\left[\phi_i\left(m_i\left(\frac{\partial}{\partial\mu_i}\left(\frac{\partial\mu_i}{\partial\eta_i}\right)^3 + a_i\right) + \frac{\partial m_i}{\partial\mu_i}\left(\frac{\partial\mu_i}{\partial\eta_i}\right)^3\right)\right.\right.$$
$$\left.\left. + w_i\frac{\partial a_i}{\partial\mu_i}\right]\frac{\partial\mu_i}{\partial\eta_i}\right\}x_{ir}x_{is}x_{it}x_{iu},$$

$$\kappa_{rst}^{(R)} = -\sum_{i=1}^{n}\left\{\phi_i\left[\left(3\phi_i m_i + \phi_i^2\left(\frac{\partial m_i}{\partial\phi_i}\right)\right)\left(\frac{\partial\mu_i}{\partial\eta_i}\right)^3 + \left(2w_i + \phi_i\left(\frac{\partial w_i}{\partial\phi_i}\right)\right)a_i\right]\left(\frac{\partial\phi_i}{\partial\zeta_i}\right)\right\}$$
$$\times x_{ir}x_{is}x_{it}z_{iR},$$

$$\kappa_{rsR}^{(t)} = \sum_{i=1}^{n}\left\{\left[2u_i\left(\frac{\partial}{\partial\mu_i}\frac{\partial\mu_i}{\partial\eta_i}\right)\left(\frac{\partial\mu_i}{\partial\eta_i}\right)^2 - c_i b_i + \left(\frac{\partial u_i}{\partial\mu_i}\right)\left(\frac{\partial\mu_i}{\partial\eta_i}\right)^3 - \left(\frac{\partial c_i}{\partial\mu_i}\right)\left(\frac{\partial}{\partial\mu_i}\frac{\partial\mu_i}{\partial\eta_i}\right)\right.\right.$$
$$\left.\left.\times\left(\frac{\partial\mu_i}{\partial\eta_i}\right)^2\right]\left(\frac{\partial\phi_i}{\partial\zeta_i}\right)\right\}x_{ir}x_{is}x_{it}z_{iR}, \quad \kappa_{rsR}^{(S)} = \sum_{i=1}^{n}\left\{\left\{\left[\left(\frac{\partial u_i}{\partial\phi_i}\right)\left(\frac{\partial\phi_i}{\partial\zeta_i}\right) + u_i\left(\frac{\partial}{\partial\phi_i}\frac{\partial\phi_i}{\partial\zeta_i}\right)\right]\right.\right.$$
$$\left.\left.\times\left(\frac{\partial\mu_i}{\partial\eta_i}\right)^2 - \left[z_i\left(\frac{\partial\phi_i}{\partial\zeta_i}\right) + c_i\left(\frac{\partial}{\partial\phi_i}\frac{\partial\phi_i}{\partial\zeta_i}\right)\right]\left(\frac{\partial}{\partial\mu_i}\frac{\partial\mu_i}{\partial\eta_i}\right)\left(\frac{\partial\mu_i}{\partial\eta_i}\right)\right\}\left(\frac{\partial\phi_i}{\partial\zeta_i}\right)\right\}$$
$$\times x_{ir}x_{is}z_{iR}z_{iS},$$

$$\kappa_{rRS}^{(s)} = \sum_{i=1}^{n}\left\{\left[-\left(\frac{\partial r_i}{\partial\mu_i}\right)\left(\frac{\partial\phi_i}{\partial\zeta_i}\right)^2 - \left[\left(\frac{\partial c_i}{\partial\mu_i}\right)\left(\frac{\partial\mu_i}{\partial\eta_i}\right) + c_i\left(\frac{\partial}{\partial\mu_i}\frac{\partial\mu_i}{\partial\eta_i}\right)\right]\left(\frac{\partial}{\partial\phi_i}\frac{\partial\phi_i}{\partial\zeta_i}\right)\right.\right.$$
$$\left.\left.\times\left(\frac{\partial\phi_i}{\partial\zeta_i}\right)\right]\left(\frac{\partial\mu_i}{\partial\eta_i}\right)\right\}x_{ir}x_{is}z_{iR}z_{iS}, \quad \kappa_{rRS}^{(T)} = -\sum_{i=1}^{n}\left\{\left(\frac{\partial r_i}{\partial\phi_i}\right)\left(\frac{\partial\phi_i}{\partial\zeta_i}\right)^3 + z_i\left(\frac{\partial\mu_i}{\partial\eta_i}\right)\right.$$
$$\left.\times\left(\frac{\partial\phi_i}{\partial\zeta_i}\right)^2\left(\frac{\partial}{\partial\phi_i}\frac{\partial\phi_i}{\partial\zeta_i}\right) + r_i\left(\frac{\partial}{\partial\phi_i}\left(\frac{\partial\phi_i}{\partial\zeta_i}\right)^2\right)\left(\frac{\partial\phi_i}{\partial\zeta_i}\right) + c_i v_i\left(\frac{\partial\mu_i}{\partial\eta_i}\right)\right\}x_{ir}z_{iR}z_{iS}z_{iT},$$

$$\kappa_{RST}^{(r)} = -\sum_{i=1}^{n}\left\{\left[\left(\frac{\partial s_i}{\partial\mu_i}\right)\left(\frac{\partial\phi_i}{\partial\zeta_i}\right)^3 + \left(\frac{\partial d_i}{\partial\mu_i}\right)\left(\frac{\partial}{\partial\phi_i}\left(\frac{\partial\phi_i}{\partial\zeta_i}\right)^3\right)\right]\left(\frac{\partial\mu_i}{\partial\eta_i}\right)\right\}x_{ir}z_{iR}z_{iS}z_{iT},$$

$$\kappa_{RST}^{(U)} = -\sum_{i=1}^{n}\left\{\left[\left(\frac{\partial s_i}{\partial\phi_i}\right)\left(\frac{\partial\phi_i}{\partial\zeta_i}\right)^3 + 2s_i\left(\frac{\partial}{\partial\phi_i}\left(\frac{\partial\phi_i}{\partial\zeta_i}\right)^3\right) + d_i\left(\frac{\partial t_i}{\partial\phi_i}\right)\right]\left(\frac{\partial\phi_i}{\partial\zeta_i}\right)\right\}$$
$$\times z_{iR}z_{iS}z_{iT}z_{iU}.$$

The second order derivatives of the log-likelihood cumulants can expressed as follows:

$$
\kappa_{rs}^{(tu)} = -\sum_{i=1}^{n}\left\{\phi_i^2\left[\phi_i\left(m_i\left(\frac{\partial}{\partial\mu_i}\left(\frac{\partial\mu_i}{\partial\eta_i}\right)^3 + \frac{2}{3}a_i\right) + \left(\frac{\partial\mu_i}{\partial\eta_i}\right)^3\frac{\partial m_i}{\partial\mu_i}\right) + \frac{2}{3}w_i\frac{\partial a_i}{\partial\mu_i}\right]\frac{\partial\mu_i}{\partial\eta_i}\right\}
$$
$$
\times x_{ir}x_{is}x_{it}x_{iu},
$$

$$
\kappa_{rs}^{(Rt)} = \sum_{i=1}^{n}\left\{\left[\left(\frac{\partial u_i}{\partial\mu_i}\right)\left(\frac{\partial\mu_i}{\partial\eta_i}\right)^2 + u_i\left(\frac{\partial}{\partial\mu_i}\left(\frac{\partial\mu_i}{\partial\eta_i}\right)^2\right)\right]\left(\frac{\partial\mu_i}{\partial\eta_i}\right)\left(\frac{\partial\phi_i}{\partial\zeta_i}\right)\right\}x_{ir}x_{is}x_{it}z_{iR},
$$

$$
\kappa_{rs}^{(RS)} = \sum_{i=1}^{n}\left\{\left[\left(\frac{\partial u_i}{\partial\phi_i}\right)\left(\frac{\partial\phi_i}{\partial\zeta_i}\right) + u_i\left(\frac{\partial}{\partial\phi_i}\frac{\partial\phi_i}{\partial\zeta_i}\right)\right]\left(\frac{\partial\mu_i}{\partial\eta_i}\right)^2\left(\frac{\partial\phi_i}{\partial\zeta_i}\right)\right\}x_{ir}x_{is}z_{iR}z_{iS},
$$

$$
\kappa_{rR}^{(st)} = \sum_{i=1}^{n}\left\{\left[-\phi_i^2\left(m_i + \left(\frac{\partial}{\partial\mu_i}\frac{\partial w_i}{\partial\phi_i}\right)\right)\left(\frac{\partial\mu_i}{\partial\eta_i}\right)^3 - \left(\frac{\partial c_i}{\partial\mu_i}\right)\left(\frac{\partial}{\partial\mu_i}\left(\frac{\partial\mu_i}{\partial\eta_i}\right)^3\right) - c_ib_i\right]\right.
$$
$$
\left.\times\left(\frac{\partial\phi_i}{\partial\zeta_i}\right)\right\}x_{ir}x_{is}x_{it}z_{iR},
$$

$$
\kappa_{rR}^{(Ss)} = \sum_{i=1}^{n}\left\{\left[-z_i\left(\frac{\partial}{\partial\mu_i}\frac{\partial\mu_i}{\partial\eta_i}\right)\left(\frac{\partial\phi_i}{\partial\zeta_i}\right)^2 - c_i\left(\frac{\partial}{\partial\mu_i}\frac{\partial\mu_i}{\partial\eta_i}\right)\left(\frac{\partial}{\partial\phi_i}\frac{\partial\phi_i}{\partial\zeta_i}\right)\left(\frac{\partial\phi_i}{\partial\zeta_i}\right) - \left(\frac{\partial z_i}{\partial\mu_i}\right)\right.\right.
$$
$$
\left.\left.\times\left(\frac{\partial\mu_i}{\partial\eta_i}\right)\left(\frac{\partial\phi_i}{\partial\zeta_i}\right)^2 - \left(\frac{\partial c_i}{\partial\mu_i}\right)\left(\frac{\partial}{\partial\phi_i}\frac{\partial\phi_i}{\partial\zeta_i}\right)\left(\frac{\partial\phi_i}{\partial\zeta_i}\right)\left(\frac{\partial\mu_i}{\partial\eta_i}\right)\right]\left(\frac{\partial\mu_i}{\partial\eta_i}\right)\right\}x_{ir}x_{is}z_{iR}z_{iS},
$$

$$
\kappa_{rR}^{(ST)} = -\sum_{i=1}^{n}\left\{z_i\left(\frac{\partial}{\partial\phi_i}\left(\frac{\partial\phi_i}{\partial\zeta_i}\right)^3\right)\left(\frac{\partial\mu_i}{\partial\eta_i}\right) + c_i\nu_i\left(\frac{\partial\mu_i}{\partial\eta_i}\right) + \left(\frac{\partial z_i}{\partial\phi_i}\right)\left(\frac{\partial\mu_i}{\partial\eta_i}\right)\left(\frac{\partial\phi_i}{\partial\zeta_i}\right)^3\right\}
$$
$$
\times x_{ir}z_{iR}z_{iS}z_{iT}, \quad \kappa_{RS}^{(rs)} = -\sum_{i=1}^{n}\left\{\left(\frac{\partial r_i}{\partial\mu_i}\right)\left(\frac{\partial\phi_i}{\partial\zeta_i}\right)^2\left(\frac{\partial\mu_i}{\partial\eta_i}\right)\right\}x_{ir}x_{is}z_{iR}z_{iS},
$$

$$
\kappa_{RS}^{(rT)} = -\sum_{i=1}^{n}\left\{\left[\left(\frac{\partial r_i}{\partial\phi_i}\right)\left(\frac{\partial\phi_i}{\partial\zeta_i}\right)^2 + r_i\left(\frac{\partial}{\partial\phi_i}\left(\frac{\partial\phi_i}{\partial\zeta_i}\right)^2\right)\right]\left(\frac{\partial\phi_i}{\partial\zeta_i}\right)\right\}x_{ir}z_{iR}z_{iS}z_{iT},
$$

$$
\kappa_{RS}^{(TU)} = -\sum_{i=1}^{n}\left\{\left[\left(\frac{\partial s_i}{\partial\phi_i}\right)\left(\frac{\partial\phi_i}{\partial\zeta_i}\right)^3 + s_i\left(\frac{\partial}{\partial\phi_i}\left(\frac{\partial\phi_i}{\partial\zeta_i}\right)^3\right) + s_i\left(\frac{2}{3}t_i\right) + d_i\left(\frac{2}{3}\frac{\partial t_i}{\partial\phi_i}\right)\right]\right.
$$
$$
\left.\times\left(\frac{\partial\phi_i}{\partial\zeta_i}\right)\right\}z_{iR}z_{iS}z_{iT}z_{iU}.
$$

## Acknowledgments

We thank two anonymous referees for comments and suggestions that led to a much improved manuscript.

## Author Contributions

**Conceptualization:** Ana C. Guedes, Francisco Cribari-Neto, Patrícia L. Espinheira.

**Data curation:** Ana C. Guedes, Francisco Cribari-Neto, Patrícia L. Espinheira.

**Formal analysis:** Ana C. Guedes, Francisco Cribari-Neto, Patrícia L. Espinheira.

**Investigation:** Ana C. Guedes, Francisco Cribari-Neto, Patrícia L. Espinheira.

**Methodology:** Ana C. Guedes, Patrícia L. Espinheira.

**Project administration:** Francisco Cribari-Neto.

**Resources:** Ana C. Guedes, Francisco Cribari-Neto, Patrícia L. Espinheira.

**Software:** Ana C. Guedes, Francisco Cribari-Neto, Patrícia L. Espinheira.

**Supervision:** Ana C. Guedes, Francisco Cribari-Neto, Patrícia L. Espinheira.

**Validation:** Ana C. Guedes, Francisco Cribari-Neto, Patrícia L. Espinheira.

**Visualization:** Ana C. Guedes, Francisco Cribari-Neto, Patrícia L. Espinheira.

**Writing – original draft:** Ana C. Guedes.

**Writing – review & editing:** Francisco Cribari-Neto, Patrícia L. Espinheira.

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
