## [Decision Letter · Decision Letter 0]

11 Apr 2021

PONE-D-21-03109

Bartlett-corrected tests for varying precision beta regressions with application to environmental biometrics

PLOS ONE

Dear Dr. Espinheira,

Thank you for submitting your manuscript to PLOS ONE. The manuscript has been assessed by two reviewers, both of whom suggest several points of improvement. We invite you to submit a revised version of the manuscript that addresses the points raised during the review process.

We look forward to receiving your revised manuscript.

Kind regards,

Ivan Kryven

Academic Editor

PLOS ONE

Journal Requirements:

In your Data Availability statement, you have not specified where the minimal data set underlying the results described in your manuscript can be found. PLOS defines a study's minimal data set as the underlying data used to reach the conclusions drawn in the manuscript and any additional data required to replicate the reported study findings in their entirety. All PLOS journals require that the minimal data set be made fully available. For more information about our data policy, please see http://journals.plos.org/plosone/s/data-availability.

Reviewers' comments:

Reviewer's Responses to Questions

**Comments to the Author**

1. Is the manuscript technically sound, and do the data support the conclusions?

Reviewer #1: Yes

Reviewer #2: Yes

2. Has the statistical analysis been performed appropriately and rigorously? 

Reviewer #1: Yes

Reviewer #2: Yes

3. Have the authors made all data underlying the findings in their manuscript fully available?

Reviewer #1: No

Reviewer #2: No

4. Is the manuscript presented in an intelligible fashion and written in standard English?

Reviewer #1: Yes

Reviewer #2: No

5. Review Comments to the Author

Reviewer #1: The work is rigorous but omits the detailed derivation of the correction terms, which, in my opinion, should be made available as a supplementary file. In addition to that I have the following comments:

1. Authors have cited many classical old work in the field. It would be better if you cite some recent work and mention to see the reference in that article. e.g. your reference [22] can be used to cite the work of Bartlett (1937) and Lawley (1956). Both the work are more than 50 years old; are fundamental contribution in Statistics and can be found in any standard book of small area estimation. Similarly, for other old work.

2. Practitioners always prefer to see the confidence interval. Bartlett corrected confidence interval of the regression coefficients under logistic regression can be found in Das, U., Dhar, S. S., & Pradhan, V. (2018). Corrected likelihood-ratio tests in logistic regression using small-sample data. Communications in Statistics-Theory and Methods, 47(17), 4272-4285. Please use the approach to construct the interval estimate of the parameters.

3. You have nuisance parameters in your model. It will be nice to have a discussion in the light of a fundamental work: Ferrari, S. L., Lucambio, F., & Cribari-Neto, F. (2005). Improved profile likelihood inference. Journal of statistical planning and inference, 134(2), 373-391. I know this is a good chunk of work, but should be mentioned about this possibility in your paper.

4. Simulation results use predictors generated from independent uniform distribution. In addition to that, i am interested to see the performance of the proposed methods when the predictors will be generated from multivariate normal distribution with different pairwise correlation structures: low correlation (say -/ + 0.1), medium correlation (say -/ + 0.5) and decaying correlation (say 0.25^|i-j|)

5. Correct the sentence in lines 186, 207 etc. and read carefully the English.

Reviewer #2: In this paper, the authors examine beta regression linear models, which include a linear model for the mean term and another linear model for the precision allowing different values for the latter in each observation. Bartlett corrections are used for the inference involving likelihood ratio tests to improve the asymptotic distribution of the statistic considering that small sample sizes are used. Three variants of these corrections are presented and compared through simulation examples. A non-simulated data set concerning intelligence and prevalence of atheism is fitted and tests are applied.

I enjoyed readying this paper. In general, it is well structured and counts a complete full story. The problems I found are mainly in terms of improving it in how it is read and to interest more the reader in what they are telling us. I found some problems concerning redaction in some paragraphs, as well as in terms of the use of English, particularly in the Introduction, which could be easily corrected by careful readying and editing the complete document. I think that one problem it has now is that it is not easy to find all the main contributions from the beginning, some are hidden in different parts of the paper. Also, there is lack of explanation of where and why certain model aspects are considered, for instance in terms of the link functions used or why having a different precision parameter in each observation is relevant.

1) In line 50 it is stated that varying precision beta regressions and corrections are commonly used. It would be useful for the reader to know in which areas they are used or general examples, including a motivation of why the precision parameter could vary. It seems that this term is somehow similar to heteroscedasticity in linear models, maybe a link with these models could be useful as a motivation. In the example, the readers analyze incidences (rates or risks), thus these beta models can be used for that type of data, but what advantages they have over Poisson models with offset? Or logistic models? (with varying or not precision terms) or in what type of data they are preferred?

2) In line 60 the example is presented. It could be useful from the beginning specifying what the response is. In general, in the paper the authors present with more detail the explanatory variable (intelligence) than the response. The response is what justifies the use of the specific distribution and this should be presented earlier. Also, I am not sure whether religious belief is an appropriate name for the variable, though I am not expert, perhaps prevalence of atheism as stated in the results is a better name.

3) In line 80, the authors talk about the “rate at which size distortions vanish”. Later in the paper, they explain what this refers to in lines 135 to 136. I think that it should be explained earlier, considering that as I understand it is one contribution in this paper. In fact, something that it is not clear is whether some of the improved test statistics are proposed by the authors for this paper or not. It seems that the authors calculated all the matricial formulas, which involved a lot of algebra and calculus, for this particular beta model, but not the statistics. The authors should be more emphatic from the introduction about their contribution, in terms of being clear that the derivation and all mathematical calculations for the model are their original work, and if they proposed any statistic for this paper or not, they should be clearer.

4) After line 84, the authors introduce the two models and though it can be inferred that they are talking about a beta distribution, I think that in terms of presenting a complete model they should state which distribution is assumed for the response. I understand that the precision parameter does not have an associated distribution, but it could seem as so, when considering how the model is presented. In a similar note, functions g_1 and g_2 are not discussed in practical terms, is it used the canonical link for g_1? Or what functions are used or allow having estimators with better statistical properties? In the simulations, some functions are used and in the example others, and the reader is a little confused of what function should be used.

5) It seems all notation in the section starting in line 101 is in terms of the beta regression model, which is fine, but the reader is not sure which results are general (which seems to be all page 4) and which ones are particular for these models (which seems to be most of page 5). Maybe if the authors were more explicit saying at the beginning that they are using for simplicity notation concerning the beta models, but that the results apply for any model, and afterwards stating which results are particular for their beta models.

6) Notation can become difficult, maybe in some parts it could be useful to remind the reader the meaning of some letters. For instance, after the formula w_{b1} (after line 109), they could say: “whose expected value equals the number of parameters of interest l”

7) In page 5, after line 116 the first and second sentence seems out of context, it seems that it belongs to a review of the literature. I think the idea here was to introduce that the Bartlett- corrected tests have been obtained in different models, and that now they are obtaining them for beta regressions. All this is fine, but maybe it is more appropriate for the Introduction. In fact, in a sentence there, they clearly state which is their main goal and contribution, which could be stated in the Introduction. Here, they could introduce a sentence saying that from there on what they present is their own work (as I understand all the previous information in this section is known and already used).

8) In lines close to those of the previous comment, the authors state that the beta and delta parameters are not orthogonal, but I was not sure why this state is true.

9) The Bartlett correction test statistics seem variants or functions of the correction c ¿why they were used as such? is one of these corrections proposed specifically for this paper? It seems that they were not.

10) For w_{a2} in line128 it is not explained the meaning or the formula associated with letter \\Xi.

11) In line 214, it could be specified which values are used. Here they use the same value for \\beta_4 and \\gamma. Would having different values affect the results?

12) In line 265 it is stated that only some countries were used for the analysis, but since you are studying Scenario 3, countries with more atheists, I think the results are probably biased to countries that already have a greater development, and care should be taken in some of the conclusions that are provided at the end of the Example.

13) In lines 268-275 it is not clear whether these variables are available in the same provided reference.

14) After the model in line 275 the authors mention an improved fit, but not what statistic was used.

15) Lines 281-282 includes an interaction, but it was not used after that, or perhaps I was not able to understand the redaction of this sentence. In fact, all sentence from line 280 to 282 should be rewritten. I think the authors wanted to say here that they tested whether \\beta_4 and \\beta_5 were zero in the model.

16) Maybe to improve the redaction from line 317, the models can be rewritten using the notation M_1 for the model fitted in reference [7] and M_2 for the model proposed by the authors.

17) As I understand, and from a point of view of medical studies, intelligence is your explanatory variable and the others are variables you want to control for, is that so? If it is, maybe it could be said.

18) Was possible multicollinearity analyzed in the example? This is an important feature that should be analyzed.

19) Lines 352-354 point out a contribution, in terms of these particular data, but still a contribution. I am not sure whether this could added to the discussion or somewhere more visible.

20) I think it could be interesting to make available, perhaps as supplementary material, the R code related to the simulations or the real data example. And by the way, is the data set fully available?

21) Something I was wondering is what happens when the response variable is heavily clustered around or is zero? Can the models still be used or modified? Can your results be generalized for any member of the exponential family? If the answer involves a new research question, it could be relevant to include in the discussion some of these aspects or others as future work and perspectives.

22) Authors should check that reference numbers, particularly for sections and subsections is correct, and that they appear in the document.

Particular comments:

Line 4, it seems it should say “of interest”

Line 9, perhaps it should be “Its density function”

Line 12 , which authors?

Line 16, “set of”

Line 107 There is something odd in how all the symbols and redaction are presented.

After line 109, there is something strange in presenting just the number to start the paragraph, maybe something like “In [11] it was proposed”

After line 120 tr() should be used instead of tr{}

Line 162. Perhaps it should say “ratio test \\omega”

Lines 186-187 have several redaction problems.

Line 200. Perhaps “consider” instead of “move to”

Line 202. Perhaps could be helpful to remind the reader of the meaning of fixed precision, for instance adding “that is, we test whether the precision is a constant”

Line 336. All redaction was strange here, I think they wanted to say that “In [7], Figure 4, the author plotted”, meaning that the authors in that reference have a Figure 4.

6. PLOS authors have the option to publish the peer review history of their article (what does this mean?). If published, this will include your full peer review and any attached files.

Reviewer #1: No

Reviewer #2: No

---

## [Author Response · Author response to Decision Letter 0]

19 May 2021

Thank you for considering the revised version of our paper for publication. 

We complied with the suggestions, corrected the identified errors and did the additional procedures requested by both the reviewers and the editor. Additionally, we answer the questions made, always trying to clarify them as much as possible.

We attach a pdf document with the details of the entire review process that we carried out following what was requested by the reviewers and editor.

---

## [Decision Letter · Decision Letter 1]

3 Jun 2021

Bartlett-corrected tests for varying precision beta regressions with application to environmental biometrics

PONE-D-21-03109R1

Dear Dr. Espinheira,

We’re pleased to inform you that your manuscript has been judged scientifically suitable for publication and will be formally accepted for publication once it meets all outstanding technical requirements. Please note that Reviewer 2 has made several minor suggestions regarding presentation style. You may incorporate them when preparing the final version for production.

Kind regards,

Ivan Kryven

Academic Editor

PLOS ONE

Additional Editor Comments (optional):

Reviewers' comments:

Reviewer's Responses to Questions

**Comments to the Author**

1. If the authors have adequately addressed your comments raised in a previous round of review and you feel that this manuscript is now acceptable for publication, you may indicate that here to bypass the “Comments to the Author” section, enter your conflict of interest statement in the “Confidential to Editor” section, and submit your "Accept" recommendation.

Reviewer #1: All comments have been addressed

Reviewer #2: All comments have been addressed

2. Is the manuscript technically sound, and do the data support the conclusions?

Reviewer #1: Yes

Reviewer #2: Yes

3. Has the statistical analysis been performed appropriately and rigorously? 

Reviewer #1: Yes

Reviewer #2: Yes

4. Have the authors made all data underlying the findings in their manuscript fully available?

Reviewer #1: (No Response)

Reviewer #2: Yes

5. Is the manuscript presented in an intelligible fashion and written in standard English?

Reviewer #1: Yes

Reviewer #2: Yes

6. Review Comments to the Author

Reviewer #1: (No Response)

Reviewer #2: Thank you very much for carefully applying all corrections, answering all my questions, and making available both data and R code. The paper has been greatly improved, now it is easier to read and understand, and all the contributions are clearer and easy to find. Technically, there are no more errors I could identify. Hence, the following lines concern a few redaction problems I identified after reading the revised version.

1) Line 67. Perhaps it should say “considered” instead of “consider”

2) Line 87. The sentence “To the best…atheists” is odd. Maybe it should be written as “To the best of our knowledge, this is the first analysis concerning the maximal impact…”

3) Line 140, after the last line in blue. Perhaps here it could be reminded to the reader what was briefly mentioned in line 65, to emphasize that this is one of the main challenges and its meaning. For instance “..are not orthogonal (Fisher information matrix is not block diagonal)…”

4) Line 156. Redaction concerning this sentence is odd. Perhaps something like “The author in [26] proposed the following… ”

5) Line 371. I found that this sentence was odd. Maybe it could say “…relative to the model presented in [13]”

6) Line 380. Perhaps, it is obvious that the letters “se” corresponds to the standard error associated with the parameters estimators, but the notation could be introduced by simply adding “standard errors, se, being…” in line 382

7) Line 405. Perhaps it is clearer this sentence if it is written as “All tests reject model M_1 at a 10% significance level; whereas the test…”

8) Line 426. This sentence could be improved by referring Figure 4, for instance “…“new” and “old” in Figure 4 refer to…”

7. PLOS authors have the option to publish the peer review history of their article (what does this mean?). If published, this will include your full peer review and any attached files.

Reviewer #1: No

Reviewer #2: No

---

## [Editor Report · Acceptance letter]

17 Jun 2021

PONE-D-21-03109R1 

Bartlett-corrected tests for varying precision beta regressions with application to environmental biometrics 

Dear Dr. Espinheira:

I'm pleased to inform you that your manuscript has been deemed suitable for publication in PLOS ONE. Congratulations! Your manuscript is now with our production department. 

Kind regards, 

on behalf of

Dr. Ivan Kryven 

Academic Editor

PLOS ONE